

# Anthropogenic and natural controls on atmospheric $\delta^{13}$C-CO$_2$
# variations in the Yangtze River Delta: Insights from a carbon
# isotope modeling framework
Cheng Hu[1, 2*], Jiaping Xu[3], Cheng Liu[4], Yan Chen[3], Dong Yang[5], Wenjing Huang[2], Lichen
Deng[6], Shoudong Liu[2], Timothy J. Griffis[7**], and Xuhui Lee [8]
[1] College of Biology and the Environment, Joint Center for sustainable Forestry in Southern China,
Nanjing Forestry University, Nanjing, 210037, China
[2] Yale-NUIST Center on Atmospheric Environment, International Joint Laboratory on Climate and
Environment Change (ILCEC), Nanjing University of Information, Science & Technology, Nanjing,
210044, China
[3] Jiangsu Climate Center, China Meteorological Administration, Jiangsu Nanjing 210009, China
[4] Jiangxi Province Key Laboratory of the Causes and Control of Atmospheric Pollution, East China
University of Technology, Nanchang 330013, China
[5] Ningbo Meteorological Observatory, Ningbo 315012, China
[6] Ecological Meteorology Center, Jiangxi Meteorological Bureau, Nanchang 330096, China
[7] Department of Soil, Water, and Climate, University of Minnesota-Twin Cities, St. Paul, Minnesota,
USA
[8] School of Forestry and Environmental Studies, Yale University, New Haven, Connecticut, USA
Correspondence:
*Cheng Hu, College of Biology and the Environment, Joint Center for sustainable Forestry in Southern
China, Nanjing Forestry University, Nanjing, 210037, China. nihaohucheng@163.com or
huxxx991@umn.edu
** Timothy J. Griffis, Department of Soil, Water, and Climate, University of Minnesota, St. Paul, MN
55108, timgriffis@umn.edu





**Abstract:**

The atmospheric $CO_2$ mixing ratio and its $\boldsymbol{\delta}^{13}C$-$CO_2$ composition contain important $CO_2$ sink and source information spanning from ecosystem to global scales. The observation and simulation for both $CO_2$ and its carbon isotope ratio ($\boldsymbol{\delta}^{13}C$-$CO_2$) can be used to constrain regional emissions and better understand the anthropogenic and natural mechanisms that control $\delta^{13}C$-$CO_2$ variations. Such work remains rare for urban environments, especially megacities. Here, we used near-continuous $CO_2$ and $\boldsymbol{\delta}^{13}C$-$CO_2$ measurements, from September 2013 to August 2015, and inverse modeling to constrain the $CO_2$ budget and investigate the main factors that dominated $\boldsymbol{\delta}^{13}C$-$CO_2$ variations for the Yangtze River Delta (YRD) region, one of the largest anthropogenic $CO_2$ hotspots and densely populated regions in China. We used the WRF-STILT model framework with category-specified EDGAR v432 $CO_2$ inventories to simulate hourly $CO_2$ mixing ratios and $\boldsymbol{\delta}^{13}C$-$CO_2$, evaluated these simulations with observations, and constrained the anthropogenic $CO_2$ emission categories. Our study shows that: (1) Top-down and bottom-up estimates of anthropogenic $CO_2$ emissions agreed well (bias < 6%) on an annual basis; (2) The WRF-STILT model performed well in reproducing the observed diel and seasonal atmospheric $\boldsymbol{\delta}^{13}C$-$CO_2$ variations; (3) Anthropogenic $CO_2$ emissions played a much larger role than ecosystems in controlling the $\boldsymbol{\delta}^{13}C$-$CO_2$ seasonality. When excluding ecosystem respiration and photosynthetic discrimination in the YRD area, $\boldsymbol{\delta}^{13}C$-$CO_2$ seasonality increased from 1.53‰ to 1.66‰; (4) Atmospheric transport processes in summer amplified the cement $CO_2$ enhancement proportions in the YRD area, which dominated monthly $\boldsymbol{\delta}$s variations. These findings support that the combination of long-term atmospheric carbon isotope observations and inverse modeling can provide a powerful constraint on the carbon cycle of these complex megacities.

**Keywords:** cements production, $^{13}C/^{12}C$ ratio, WRF-STILT model, plants photosynthetic discrimination



## 1. Introduction

Urban landscapes account for 70% of global $CO_2$ emissions and represent less than 1% of Earth's land area (Seto et al., 2014). Such $CO_2$ hotspots play a dominant role in controlling the rise in atmospheric $CO_2$ concentrations, which exceeded 412 ppm in December 2019 for global monthly average observations (https://www.esrl.noaa.gov/gmd/ccgg/trends/). Furthermore, the carbon isotope ratio of $CO_2$ (i.e. $\delta^{13}C$ = $^{13}C/^{12}C$ ratio in delta notation) at the representative Mauna Loa site, USA, has steadily decreased to around -8.5‰, in December 2019 (https://www.esrl.noaa.gov/). Anthropogenic $CO_2$ emission is produced by fossil fuel burning and cement production. As the urban population is expected to increase by 2.5 to 6 billion people in 2050, anthropogenic $CO_2$ emissions are projected to increase dramatically, especially in developing regions and countries (Sargent et al., 2018; Ribeiro et al., 2019). Under such a scenario, the observations of atmospheric $CO_2$ and $\delta^{13}C$-$CO_2$ in urban landscapes are of great importance to monitoring these potential $CO_2$ emissions hotspots (Lauvaux et al., 2016; Nathan et al., 2018; Graven et al., 2018; Pillai et al., 2016; Staufer et al., 2016).

Countries are required to report their $CO_2$ emissions according to the Intergovernmental Panel on Climate Change guidelines (IPCC; e.g. IPCC 2013), and many "bottom-up" methods have long been used to estimate $CO_2$ emissions worldwide, but such methods have high uncertainties for $CO_2$ emissions at regional (20%) to city (50 to 250%) scales (Gately & Hutyra, 2017; Gately et al., 2015). These significant uncertainties are propagated into the inversion of global biological $CO_2$ flux (Zhang et al., 2014; Jiang et al., 2014; Thompson et al., 2016). By using $CO_2$ observations, the "top-down" atmospheric inversion approach is a useful tool to evaluate "bottom-up" inventories (Graven et al., 2018; L. Hu et al., 2019; Lauvaux et al., 2016; Nathan et al., 2018). Previous research has shown that additional information, such as data on atmospheric $\Delta^{14}CO_2$-$CO_2$, $\delta^{13}C$-$CO_2$, and CO, is needed to better distinguish $CO_2$ emissions from different sources and to assess their uncertainties (Chen et al., 2017; Graven et al., 2018; Nathan et al., 2018; Cui et al., 2019). The use of hourly $\delta^{13}C$-$CO_2$ observation in urban areas remains rare in inversion studies, yet such observations contain invaluable information of anthropogenic $CO_2$ from different categories.

Traditional estimates of $\delta^{13}C$-$CO_2$ using isotope ratio mass spectrometry (IRMS) are very limited because flask air sample collection requires long preparation time and is expensive. Consequently, there is a lack of high temporal and long-term observations of $\delta^{13}C$-$CO_2$ (Sturm et al., 2006). Isotope ratio infrared spectroscopy technology (IRIS) has overcome these limitations. As a result, *in situ* air sample analyses using IRIS analyzers are resulting in dense time series of $\delta^{13}C$-$CO_2$. However, most of the established




long-term IRMS and IRIS $\delta^{13}$C-$CO_2$ measurement sites are representative of "background" or natural
ecosystem conditions at locations far away from urban landscapes (Chen et al., 2017; Griffis, 2013).
To date, long-term (> 1 year) and continuous observations of both $CO_2$ and $\delta^{13}$C-$CO_2$ have been reported
for only five cities, including Bern, Switzerland (Sturm et al., 2006); Boston, USA (McManus et al.,
2010); Salt Lake City, USA (Pataki et al., 2006); Beijing, China (Pang et al., 2016); and Nanjing, China
(Xu et al., 2017). In these previous investigations, significant diel and seasonal variations of $\delta^{13}$C-$CO_2$
have been observed; these patterns were modulated by fossil fuel combustion, plant respiration and
photosynthesis, and changes in the height of the atmospheric boundary layer (Sturm et al., 2006; Guha
and Ghosh, 2010). No study has quantified the impact of each factor on the seasonal variation of $\delta^{13}$C-
$CO_2$. This represents an important knowledge gap in understanding the underlying mechanisms of carbon
cycling in complex urban ecosystems.
The traditional $\delta^{13}$C-$CO_2$ isotope partitioning methods (including Miller-Tans and the Keeling plot
approaches) have be used to constrain different $CO_2$ sources worldwide (Keeling, 1960; Vardag et al.,
2015; Newman et al., 2016; Pang et al., 2016; Xu et al., 2017). These methods are based on the
assumption that partitioned atmospheric $CO_2$ enhancement components from different sources can
represent $CO_2$ emissions at the "target area" (Miller and Tans, 2003; Ballantyne et al., 2011). Carbon
dioxide emissions are highly inhomogeneous at the urban scale, with extremely strong point/line sources,
and the final partitioning results are highly uncertain without considerations of source footprint
characteristics (Gately & Hutyra, 2017; Cui et al., 2019; Martin et al., 2019). Atmospheric transport
models can help to resolve such problems, and the coupling of atmospheric transport models with isotope
observations have recently be applied in global and regional $CO_2$ partitioning studies (Chen et al., 2017;
Cui et al., 2019; Graven et al., 2018; C. Hu et al., 2018b). Although urban $CO_2$ inversion has been applied
successfully in several studies in Europe and the United States (Bréon et al., 2015; Turnbull et al., 2015;
Pillai et al., 2016; Brioude et al., 2013; Turner et al., 2016),  urban $CO_2$  inversions in China are rare
(Berezin et al., 2013; C. Hu, 2018a; Worden et al., 2012), presumably because of the scarcity of high
quality $\delta^{13}$C-$CO_2$ and $CO_2$ observations.
The Yangtze River Delta (YRD) ranks as one of the most densely populated regions in the world and is
an important anthropogenic $CO_2$ hotspot. Major anthropogenic sources include power industry, oil
refineries/transformation and cement productions. Having the largest source of cement-derived $CO_2$
production across China and the world (Cai et al., 2015), the YRD contributed 20% of national cement
production, nearly 12% of world's total cement output in 2014 (Xu et al., 2017; Yang et al., 2017).
Besides the anthropogenic factors, natural ecosystems and croplands act as significant $CO_2$ sinks and
sources within the YRD. Independent quantification of the fossil and cement $CO_2$ emission and



assessment of their impact on atmospheric $\delta^{13}$C-CO$_2$ have potential to improve our understanding of
urban CO$_2$ cycling. Further, the observations and simulations of both atmospheric CO$_2$ and $\delta^{13}$C-CO$_2$ can
help us relate atmospheric CO$_2$ dynamics with future emissions control strategies.
Here, we combine long-term (>2 years) CO$_2$ and $\delta^{13}$C-CO$_2$ observations with atmospheric transport
model simulations to study urban atmospheric CO$_2$ and $\delta^{13}$C-CO$_2$ variations. The objectives were to: (1)
Constrain anthropogenic CO$_2$ emissions and determine the main sources of uncertainty for $\delta^{13}$C-CO$_2$
simulations, and (2) Quantify the relative contributions of each factor (i.e. background, anthropogenic
CO$_2$ emissions especially for cement production, ecosystem photosynthesis and respiration) to seasonal
variations of atmospheric $\delta^{13}$C-CO$_2$.

## 144    2. Materials and methods

### 145    2.1 Observations of atmospheric CO$_2$ mixing ratio, $\delta^{13}$C-CO$_2$ and supporting variables

The observation site is located on the Nanjing University of Information Science and Technology campus
(hereafter NUIST, 32$^o$12'N, 118$^o$43'E, green dot in Figure 1a). Continuous atmospheric CO$_2$ mixing
ratios and $\delta^{13}$C-CO$_2$ were measured at a height of 34 m above ground with an IRIS analyzer (model
G1101-i, Picarro Inc., Sunnyvale, CA). The observation period extended from September 2013 to August
2015. Calibrations for CO$_2$ mixing ratio and $\delta^{13}$C-CO$_2$ were conducted with standard gases traceable to
NOAA-ESRL (National Oceanic and Atmospheric Administration, Earth System Research Laboratory)
standards. Calibration details are provided by Xu et al. (2017). Based on Allan variance analyses, the
hourly precisions of CO$_2$ and $\delta^{13}$C-CO$_2$ were 0.07 ppm and 0.05‰, respectively.
We separated the two-year study period into seasons (autumn: September, October, November; winter:
December, January, February; spring: March, April, May; summer: June, July, August). Further, for an
annual comparison, we examined the period from September 2013 to August 2014 (Year 2014) versus
September 2014 to August 2015 (Year 2015).
The YRD is a cement production hotspot in China (Figure 1b). It had a total population of 190 million in
2018 (Figure 2a) with 24.2 million in the city of Shanghai, 9.8 million in Hangzhou city (provincial
capital of Zhejiang), 8.4 million in Nanjing city (provincial capital of Jiangsu), and 8.1 million in Hefei
city (provincial capital of Anhui). The CO$_2$ related production data (i.e. cement) and energy consumption
data (i.e. coal and natural gas) were obtained from local official sources using the same method described
in Shen et al. (2014).
To examine the effects of plant photosynthesis on atmospheric CO$_2$ variations, we used NDVI
(Normalized Difference Vegetation Index), SIF (solar-induced chlorophyll fluorescence) and GPP (gross





primary productivity) information. These three products have a global distribution with spatial resolution
of 0.05° by 0.05°. The NDVI has a temporal resolution of 16 days and SIF and GPP products have a
temporal resolution of 8 days (Li & Xiao, 2019; http://globalecology.unh.edu/data/). Land-use and land-
cover classification in Yangtze River Delta for 2014 was applied by using NDVI data of MOD13A2.
**2.2 Simulation of atmospheric $\delta^{13}$C-CO$_2$**
**2.2.1 General equations**
The simulation of atmospheric $\delta^{13}$C-CO$_2$ is based on mass conservation. First, we briefly describe the
simulation of atmospheric CO$_2$ mixing ratios (more details are provided in Section 2.2.2), following the
previous work of Hu et al., (2018b), where CO$_2$ was simulated as the sum of background (CO$_{2\_bg}$) and the
contribution from all regional sources/sinks ($\Delta$CO$_2$), as
$$CO_{2\_ms} = CO_{2\_bg} + \Delta CO_2 \qquad (1)$$

Based on mass conservation, we estimated the $^{13}$CO$_2$ composition by multiplying the left and right hands
of equation (1) by $\delta^{13}$C,
$$\delta^{13}C_a = \frac{\delta^{13}C_{bg} \times CO_{2\_bg} + \sum_{i=1}^{n} \delta_i^{13} \times [\Delta CO_2]_i}{CO_{2\_ms}} \qquad (2)$$

where $\delta^{13}C_a$ and $\delta^{13}C_{bg}$ represent the atmospheric $\delta^{13}$C-CO$_2$ and background $\delta^{13}$CO$_2$, $\delta_i^{13}$ is the $\delta^{13}$C-CO$_2$
for end-member $i$ (including anthropogenic and biological source categories). The $\delta^{13}$C-CO$_2$ contributions
from all regional sources/sinks can be further reformatted as equation 3,
$$\sum_{i=1}^{n} \delta_i^{13} \times [\Delta CO_2]_i = \delta_s \times \Delta CO_2 \qquad (3)$$

where $\delta_s$ is the mixture of all regional end-members (Newman et al., 2008), which will be described in
detail in section 2.2.5, and $\Delta$CO$_2$ represents the sum of CO$_2$ mixing ratio from all regional contributions
(hereafter  total CO$_2$ enhancement). The product of $\delta_s \times \Delta$CO$_2$ can be treated as the regional source term.
To date, there are no available global $\delta^{13}$C-CO$_2$ background products and the choice of $\delta^{13}C_{bg}$ is essential
to simulating $\delta^{13}C_a$. Here, we apply three strategies. First, we used discrete $\delta^{13}$C-CO$_2$ flask observations
at Mount Waliguan (hereafter WLG, 36°17'N, 100°54'E; https://www.esrl.noaa.gov/gmd/dv/data/) to
represent the $\delta^{13}$C-CO$_2$ background signal at our site. These observations were measured at weekly
intervals to the end of 2015. A digital filtering curve fitting (CCGCRV) regression method was applied to





derive hourly background values following Thoning et al. (1989). There are, however, reasons why WLG
may not be an ideal background site for our study domain. For example, based on the previous simulation
results for the $CO_2$ background sources, background air masses should originate from the free atmosphere
at heights of 1000 m or higher above the ground (Hu et al., 2019). Here, the WLG observations were
made near the surface. Further, WLG is not located at the border of our simulation domain 1. Therefore,
the strong vertical $\boldsymbol{\delta}^{13}$C-$CO_2$ gradients between the boundary layer and the free tropospheric atmosphere
(Chen et al., 2006; Guha et al., 2010; Sturm et al., 2013) can cause a high bias in the $\boldsymbol{\delta}^{13}$C-$CO_2$
background when using this approach.
In the second approach, the $\boldsymbol{\delta}^{13}$C-$CO_2$ background signal was estimated with wintertime "clean" air $CO_2$
and $\boldsymbol{\delta}^{13}$C-$CO_2$ observations at the NUIST site, using the following equation
$$\delta^{13}C_{bg} = \frac{\delta^{13}C_a \times [CO_2] - \sum_{i=1}^{n} \delta_i^{13} \times [\Delta CO_2]_i}{CO_{2\_bg}} \qquad (4)$$

where $\boldsymbol{\delta}^{13}$C$_a$ and [$CO_2$] represent atmospheric $\boldsymbol{\delta}^{13}$C-$CO_2$ and $CO_2$ observations at the NUIST site under
clean conditions. [$\boldsymbol{\Delta}CO_2$]$_i$ is the simulated category-specified $CO_2$ enhancement. Here, we defined clean
conditions as the lowest 5% quintile wintertime $CO_2$ observations to minimize simulated $CO_2$
enhancement errors on $\boldsymbol{\delta}^{13}$C-$CO_2$ background calculation. The $CO_{2\_bg}$ is obtained from heights 1000 m
above ground level (see Section 2.2.3).
In the third approach, we avoid the use of modeled [$\boldsymbol{\Delta}CO_2$]$_i$ results and replaced the regional source term
in equation 4 with $\boldsymbol{\delta}$s×$\boldsymbol{\Delta}CO_2$, as described in equations 3, and used the Miller-Tans regression method to
calculate monthly $\boldsymbol{\delta}$s. This approach does not require simulation of [$\boldsymbol{\Delta}CO_2$]$_i$ or the corresponding $\boldsymbol{\delta}^{13}$C-
$CO_2$ signals. The hourly $\boldsymbol{\delta}^{13}$C-$CO_2$ background value can be derived by using $\boldsymbol{\delta}$s, $CO_2$ background,
observed atmospheric $\boldsymbol{\delta}^{13}$C$_a$ and $CO_2$ (see details in Section 2.3 and supplement materials). Comparison of
these three strategies will be evaluated and discussed in Section 3.2.1. Similar methods used to derive
other background tracers have been used including $CO_2$ (Alden et al., 2016; Verhulst et al., 2017), CO
(Wang et al., 2010; Ruckstuhl et al., 2012) and $CH_4$ (Zhao et al., 2009; Verhulst et al., 2017; Hu et al.,
2019). To analyze the controlling factors for the $\boldsymbol{\delta}^{13}$C-$CO_2$ seasonality, the CCGCRV regression was
applied to the background, observations, and simulations. Finally, we derived CCGCRV curving fitting
lines and defined the difference between peak and trough in one year as the seasonality of $\boldsymbol{\delta}^{13}$C-$CO_2$.
**2.2.2 Simulation of atmospheric $CO_2$ mixing ratios**



In equation 1, the $CO_{2\_bg}$ is obtained from the Carbon Tracker 2016 product, which provides global $CO_2$
distributions from the ground level up to a height of 50 km. We used the concentration at a height of 1000
m above ground where the air mass enters study domain 1 (Figure 1a). The variable $\Delta CO_2$ was derived by
multiplying the simulated hourly footprint function with the $CO_2$ fluxes (see details in Sect. 2.2.4). The
$CO_2$ fluxes contain anthropogenic $CO_2$ emissions, biological $CO_2$ flux and biomass burning. Here the
anthropogenic $CO_2$ emission sources include power industry, combustion for manufacturing, non-metallic
minerals production (cement), oil refineries/transformation industry, energy for building and road
transportation. Theoretically, $\Delta CO_2$ represents the $CO_2$ changes contributed by every pixel within the
simulated domain. As shown by Hu et al. (2018a), most of the $\Delta CO_2$ is contributed by sink/source
activity within the YRD area. In order to quantify the relative contributions within the YRD area, we
separated the study domain into 5 zones based on provincial administrative boundaries including Jiangsu,
Anhui, Zhejiang, Shanghai, and the remaining area outside the YRD. The modeled $CO_2$ was calculated as
follows:
$$\Delta CO_2 = \sum_{i=1}^{n} flux_i \times footprint$$

(5)

where $flux_i$ corresponds to each $CO_2$ flux category simulated for each domain, and footprint is the model
simulated sensitivity of observed $CO_2$ enhancement to flux changes in each pixel as described below.

**2.2.3 WRF-STILT model configuration**

The Stochastic Time-Inverted Lagrangian Transport (hereafter STILT) model was used to generate the
above footprint, which is defined as the sensitivity of atmospheric $CO_2$ enhancement to the upwind flux at
the receptor site (observation site). The meteorological fields used to drive the STILT model were
simulated with the Weather Research and Forecasting Model (WRF3.5) at high spatial and temporal
resolutions. The innermost nested domain (D3, 3 km × 3 km, Figure 1) contains the YRD area, where the
most sensitive footprint is located, and the intermediate domain (D2, 9 km × 9 km) and outermost (D1, 27
km × 27 km) represent East China and Central and Eastern China, respectively. The WRF setup used
physical schemes and parameters that have been used previously for inverse analyses (Hu et al., 2019).
These previous studies at the NUIST observation site have shown very good performance in simulating
the meteorological fields, which is essential for reliable STILT simulations. The hourly footprint was
simulated by releasing 500 particles from the NUIST measurement site and tracking their locations every
5 minutes for a period of 7 days. Particle numbers and their residence time within half of the planetary
boundary layer (hereafter PBL) height were used to calculate the footprint over the 7 day period. For the
$CO_2$ background of each hour, we tracked the sources of air particles back trajectory at the end of 7 days



at the heights above 1000 m, and defined these $CO_2$ mixing ratios in Carbon Tracker as the hourly $CO_2$
background values (Peters et al., 2007).

### 2.2.4 A priori anthropogenic $CO_2$ emissions and net ecosystem exchange

The Emission Database for Global Atmospheric Research (EDGAR) inventory was selected as the *a*
*priori* anthropogenic $CO_2$ emissions (Figure 2a), which is based on the International Energy Agency's
(IEA) energy budget statistics and provides detailed $CO_2$ source maps (19 categories, including both
organic and fossil emissions, IEA, 2012) with global coverage at high spatial resolution ($0.1^o \times 0.1^o$). The
EDGAR $CO_2$ emissions are the most up-to-date global inventory (Janssens-Maenhout et al., 2017;
Schneising et al., 2013). Other inventories, including the Fossil Fuel Data Assimilation System (FFDAS,
Rayner et al., 2010) and the Open-source Data Inventory for Anthropogenic $CO_2$ (ODIAC, Oda et al.,
2018) also provide global $CO_2$ emissions. However, these inventories only provide total $CO_2$ emissions or
have very limited emission categories, which limit our ability to provide isotope end-member information.
EDGAR v432 provides emission estimates at a monthly time scale. Here, we applied hourly scaling
factors for different categories following Hu et al., (2018a). EDGAR v432 is available only for 2010. We
assume that each $CO_2$ category changes linearly from its 2010 value (Peters et al., 2007) and apply an
annual scaling factor of 1.145 to derive $CO_2$ emissions for 2014 and 2015. This scaling factor is based on
Carbon Tracker anthropogenic $CO_2$ emissions for YRD.
The biological flux or net ecosystem $CO_2$ exchange (NEE) and biomass burning $CO_2$ emissions come
from Carbon Tracker *posteriori* flux at 3-hour intervals and at a spatial resolution of $1^o \times 1^o$. Because
NEE is much smaller than the anthropogenic $CO_2$ emissions in such densely developed urban landscapes,
we homogeneously distributed this flux at a spatial resolution of $0.1^o$ within each grid to match the
footprint.

### 2.2.5 The simulation of carbon isotope ratio of all sources ($\delta$s)

The carbon isotope ratio of all the surface sources was calculated as (Newman et al., 2008):
$$\sum_{i=1}^{n} \delta_i \times p_i = \delta_s$$
(6)

where $\delta_i$ is the $\delta^{13}$C-$CO_2$ value from source category *i*, and $p_i$ is the corresponding enhancement
proportion. Based on fossil fuel usage characteristics in YRD, we reassigned the EDGAR v432 categories
according to fuel types. Coal was the fuel type for manufacturing, oil for oil refinery, natural gas for
buildings, and diesel and gasoline for transportation. The power industry consumed 5% natural gas and 95%





coal based on local activity data in YRD (China statistical Yearbook, 2015). The non-metallic mineral
production was mainly for cement. Chemical processes were mainly ammonia synthesis. Based on a
literature review and our previous work (Xu et al., 2017), typical $\delta^{13}$C-$CO_2$ values for natural gas (−39.06‰
± 1.07‰), coal (−25.46‰ ± 0.39‰), fuel oil (−29.32‰ ± 0.15‰), gasoline (−28.69‰ ± 0.50‰),
ammonia synthesis (−28.18‰ ± 0.55‰), and diesel (−28.93‰ ± 0.26‰), pig iron (−24.90‰ ± 0.40‰),
crude steel (−25.28‰ ± 0.40‰), cement (0‰ ± 0.30‰), biological and organic emissions (−28.20‰ ±
1.00‰) were used in this study. We also applied a value of −28.20‰ for photosynthesis (Griffis et al.,
2008; Lai et al., 2014) because YRD is a region dominated by $C_3$ plants.
To evaluate the simulated $\delta_s$, we applied the Miller-Tans and Keeling plot approaches to derive $\delta_s$ from
the observed concentration and atmospheric $^{13}CO_2$-$CO_2$ (Xu et al. 2017). We then used the results to
evaluate the calculations made with Equation (6).

**2.3 Independent IPCC method for anthropogenic $CO_2$ emissions**

Large differences between different inventories have been previously found even for the same region
(Berezin et al., 2013; Andrew, 2019). For comparison with the EDGAR v432 inventory results, we
derived the anthropogenic $CO_2$ emissions by using an independent IPCC method. Here, we illustrate the
calculation for cement $CO_2$ emissions. Note that the IPCC only recommended an EF for clinker, which is
an intermediate product of cement. To calculate cement $CO_2$ emissions, we need to calculate it based on
clinker production, as shown in Equation (7),

$$CO_2[cement] = M_{cement} \times C_{clinker} \times EF_{clinker}$$

(7)

where $CO_2$[cement] is the chemical process $CO_2$ emissions for cement production, $M_{cement}$ is the
production of cement, $C_{clinker}$ represents the clinker to cement ratio (%), and $EF_{clinker}$ is the $CO_2$ emission
factor for clinker production. The IPCC recommended an $EF_{clinker}$ value of 0.52 ± 0.01 tonne $CO_2$ per
tonne clinker produced, where CaO content for clinker is assumed to be 65% with 100% CaO from
calcium carbonate material (IPCC 2013). The EF appears to be well constrained, showing little variation
among provinces with mean values ranging from 0.512 to 0.525 (Yang et al., 2017). For the $C_{clinker}$ values,
it generally showed a decreasing trend from 64.5% in 2004 to 56.9% in 2015 for all of China (Figure S1),
with an average value of 57.0% during 2014 and 2015.

**2.4 Multiplicative scaling factor method**

To quantify anthropogenic $CO_2$ emissions and to compare it with EDGAR products, we first derived the
monthly scaling factors for anthropogenic $CO_2$ emissions using a multiplicative scaling factor (hereafter





MSF) method (Hu et al., 2019; Sargent et al., 2018; He et al., 2020), and then obtained annual averages.
The monthly scaling factors (SFs) were calculated as:
$$MSF = \frac{CO_{2\_obs} - CO_{2\_bg} - \Delta CO_{2\_bio} - \Delta CO_{2\_fire}}{\Delta CO_{2\_ms}}$$
(8)

where $CO_{2\_obs}$, $\Delta CO_{2\_bio}$, $\Delta CO_{2\_fire}$ and $\Delta CO_{2\_ms}$ represent observed $CO_2$ mixing ratios, simulated $CO_2$
enhancements contributed by biological flux, biomass burning and anthropogenic emissions, respectively.
Uncertainties of all factors on the final MSFs were calculated based on Monte Carlo methods, where the
normal sample probability distribution was applied and the upper 97.5% and lower 2.5% of the values
was considered as the uncertainty for MSF (Cao et al., 2016).
**3. Results and Discussion**
**3.1 Evaluation of hourly $CO_2$ mixing ratios**
**3.1.1 Hourly and monthly $CO_2$ mixing ratio comparisons**
This section examines the general performance of simulating hourly $CO_2$ mixing ratios. The two-year
average hourly footprint is shown in Figure 2b where the source area (blue) indicates strong sensitivity of
the $CO_2$ observations to regional sources. This footprint shape is representative of the YRD area. To
quantify the relative contributions from each province, we calculated $CO_2$ enhancements contributed by
Anhui, Jiangsu, Zhejiang, Shanghai, and the remaining area outside of the YRD, respectively. The results
indicate that Jiangsu contributed approximately 80% of the total enhancement (discussed further in
Section 3.1.2). Comparisons between simulated and observed hourly $CO_2$ mixing ratios are displayed in
Figure 3a for both years. For all hourly data in each year, the model versus observation correlation
coefficient (R) was $R = 0.38$ (n = 8204, P < 0.001) and RMSE = 29.44 ppm for 2014, and $R = 0.35$ (n =
7262, P < 0.001) and RMSE = 30.22 ppm for 2015. These results indicate that the model can simulate the
synoptic and diel $CO_2$ variations over the two-year period. The model also performed well in simulating
the monthly and seasonal variations of $CO_2$ mixing ratios (daily averages are shown in Figure S2). The
simulations captured the trend of rising $CO_2$ mixing ratios after October and the drawdown of $CO_2$ below
the background value during the summer.
Figures 3b-d illustrate the average monthly daily, nighttime, and daytime $CO_2$ mixing ratios. These
monthly values contain the effects including atmospheric transport, background fields and variations in
$CO_2$ emissions. The observed and simulated $CO_2$ mixing ratios showed a significant increase from
September 2013 to January 2014. Here, the $CO_2$ mixing ratios increased by 16.0 ppm according to the
model results and 17.2 ppm according to the observations. The background values increased by 8.1 ppm
and accounted for 47% of the total $CO_2$ increase, and the net $CO_2$ flux (*a priori*) for YRD increased by



15%. We attributed the remaining 38% increase to changes in atmospheric transport processes including
lower PBL heights in January 2014 than in September 2013. To quantify how variations in PBL height
affected $CO_2$ mixing ratios, we compared the simulated monthly anthropogenic $CO_2$ enhancement
differences in the same months of different years, to eliminate the influence of monthly emission
variations on $CO_2$ enhancements. Twelve monthly paired values were used and are shown in Figure 4a.
This analysis indicates that atmospheric $CO_2$ mixing ratios decreased by about 3.7 ppm for an increase of
PBL height by 100 m.
On an annual timescale, the simulated average $CO_2$ mixing ratios were 436.63 ppm and 437.11 ppm for
2014 and 2015, respectively. Since the anthropogenic $CO_2$ emissions used in the model are the same for
both years, the simulated annual average $CO_2$ difference can be used to quantify the influence associated
with meteorological factors and ecosystem carbon cycling. Between these two years, the $CO_2$ background
increased by 1.78 ppm, the biological enhancement decreased by 1.04 ppm from 2014 to 2015. The
remaining 0.26 ppm change between 2014 and 2015 indicates a relatively small meteorological effect,
such as a slight change in dominant wind direction or a PBL height difference.
The simulated annual average NEE $CO_2$ enhancements were 2.64 ppm and 1.60 ppm for the respective
years. For comparison, the annual average anthropogenic enhancements were 36.20 ppm and 34.90 ppm
for 2014 and 2015, respectively. The monthly NEE enhancement varied from -0.1 ppm in May 2015 to
+6.0 ppm July 2014, indicating NEE contributes positively for enhancement in most months (Figure 5a),
even though the sign of monthly averaged NEE flux in summer was negative (sinks). This positive
contribution was mainly caused by diel PBL height variations between daytime (smaller negative
enhancement) and nighttime (larger positive enhancement). To further evaluate the impact of plant
photosynthetic activity on the regional $CO_2$ cycle, we examined the NDVI, SIF and GPP seasonal patterns
(Figures 4b-c). These three datasets revealed two peaks during each year, which is related to increased
photosynthetic activity. The first peak occurred in May and the second in August-September,
corresponding to the growing season of wheat and corn/rice, respectively (Deng et al., 2015). The land-
use classification in YRD for 2014 (Figure S3) shows that north YRD is dominated by agricultural land
and south dominated by forest land, and our observation site was more surrounded by agricultural land
which corresponded well with observed NDVI, SIF and GPP seasonal patterns. The peak SIF and GPP
signals during the summer were about 20 times greater than during the winter. Consequently, we can
ignore the potential influence of photosynthetic activity on the regional $CO_2$ enhancements during the
non-growing seasons.
**3.1.2 Components of urban $CO_2$ enhancement**





Here, we diagnose the source contributions to the urban $CO_2$ enhancement. The observed anthropogenic
$CO_2$ enhancements, which were derived by subtracting $CO_2$ background and simulated biological
enhancement from $CO_2$ concentration observations, were 38.36 ppm and 37.89 ppm for 2014 and 2015,
respectively. The corresponding simulated anthropogenic $CO_2$ enhancements were 36.20 ppm and 34.90
ppm. In comparison with the simulated biological $CO_2$ enhancements displayed in Figure 5a, both the
observed and simulated $CO_2$ enhancements are indicative of a large anthropogenic (fossil fuel and cement
production) $CO_2$ emission from the YRD.
Previous studies have also investigated urban $CO_2$ enhancements from a relatively broad range of
developed environments worldwide.  Verhulst et al. (2017) measured $CO_2$ mixing ratios at seven sites in
Los Angeles, USA and concluded that the mean annual enhancement varied between 2.0 ppm and 30.8
ppm, which is considerably lower than our findings. Another study in Washington, USA in February and
July 2013 showed that the $CO_2$ enhancement was less than 20 ppm (Mueller et al., 2018). The urban $CO_2$
observations and modeling study by Martin et al. (2019) at three urban sites in Eastern USA showed an
enhancement of ~21 ppm in February 2013, substantially lower (by ~20 ppm) than our observations.  The
measurements at an urban-industrial complex site in Rotterdam, Netherlands, indicated a $CO_2$
enhancement of only 11 ppm for October to December 2014 (Super et al., 2017). Our enhancements were
significantly higher than all of these previous reports, indicating greater anthropogenic $CO_2$ emissions
than other urban areas.
The anthropogenic components and source area contributions are displayed in Figure 5b-c. During the
study period the average anthropogenic enhancements were 5.1%, 80.2%, 1.9%, 4.4%, and 8.5% for
Anhui, Jiangsu, Zhejiang, Shanghai, and the remaining area outside the YRD, respectively. Although
Shanghai's area is the smallest within the YRD region and relatively distant (~300 km) from our
observation site, its maximum source contribution at times exceeded 50% (i.e. on 19[th] September 2013)
*via* long-distance transport. In general, power industry, manufacturing, non-metallic mineral production,
oil refinery, and other source categories contributed 41.0%, 21.9%, 9.3%, 11.5%, and 16.3% to the total
anthropogenic $CO_2$ enhancement, respectively. The proportions of corresponding $CO_2$ emission
categories to the total anthropogenic emissions of the YRD were 39.8%, 28.4%, 7.4%, 4.1%, and 24.4%,
respectively. We found a relatively large difference between the enhancement proportion and the
emission proportion for oil refinery (from 11.5% to 4.1%) as compared to other categories. This may be
because power industry, manufacturing and non-metallic mineral production were more homogeneously
distributed than oil refinery, and oil refinery activities were closer to our $CO_2$ observation site.
**3.1.3 Constraints on monthly anthropogenic $CO_2$ emissions**



To provide a robust comparison of bottom-up $CO_2$ emissions for YRD, we calculated anthropogenic $CO_2$
emissions from both EDGAR v432 and with activity data provided by local governments (Table 1) and
the default IPCC emission factors (https://www.ipcc-nggip.iges.or.jp/EFDB/). The total anthropogenic
$CO_2$ emissions in 2014 were $2.44 \times 10^{12}$ kg and $2.35 \times 10^{12}$ kg according to our own inventory and
EDGAR v432 $CO_2$, respectively, indicating excellent agreement (within 4%) between these approaches.
We constrained the monthly anthropogenic $CO_2$ emissions by using the MSF method (equation 8) and
computed the 12-month average to represent the years of 2014 and 2015. The *posteriori* results indicate
that the annual scaling factors were $1.03 \pm 0.10$ for 2014 and $1.06 \pm 0.09$ for 2015. The anthropogenic
$CO_2$ emissions in year 2015 did not show a significant change compared to 2014, and the overall
estimates were within the uncertainty of the estimates. After applying the average scaling factors for 2014
and 2015, the *posteriori* anthropogenic $CO_2$ emissions were $2.46 (\pm 0.24) \times 10^{12}$ kg for YRD area. The
application of the MSF method provides an overall constraint on the anthropogenic $CO_2$ emissions. As
noted, cement $CO_2$ emissions in the YRD is the largest regional source for global cement production (also
displayed in Table 1).
**3.2 Simulation of atmospheric $\delta^{13}$C-$CO_2$**
**3.2.1 Background atmospheric $\delta^{13}$C-$CO_2$**
To obtain the best representative $\delta^{13}$C-$CO_2$ background value for the study domain we examined the
values from the three strategies described above (Figure 6). We also compared the $\delta^{13}$C-$CO_2$ at the WLG
background site with observations at NUIST during winters (Figure S4). This was performed to help
simplify the comparison by removing the effects of plant photosynthetic discrimination. The $\delta^{13}$C-$CO_2$ at
the WLG site was relatively more depleted in the heavy carbon isotope (or negative, by up to 0.5‰) than
that observed at NUIST for many periods. Theoretically, there are two key factors that can cause the
urban atmospheric $\delta^{13}$C-$CO_2$ to be relatively more enriched in the heavy carbon isotope (or positive)
compared to the background values including: 1) Discrimination associated with ecosystem
photosynthesis; and 2) Discrimination associated with the $CO_2$ derived from cement production. As
shown earlier, the biological $CO_2$ enhancement was positive in winter, which implies a negligible role of
plants photosynthesis. Further, sensitivity tests for cement $CO_2$ sources showed its influence is much
smaller than observed difference in Figure S4 (discussed in section 3.3.3). Based on the above analyses
and methods introduced in Section 2.3, we concluded that WLG $\delta^{13}$C-$CO_2$ is not an ideal choice for the
domain. The wintertime $\delta^{13}$C-$CO_2$ background values, based on strategy 2, were -7.78‰ and -7.61‰ for
2013-2014 and 2014-2015, respectively. The corresponding values, based on strategy 3, were -7.70‰ and
-7.53‰. These background values are more enriched compared to the WLG observations by 0.80‰ to
1.01‰. These derived backgrounds agree well with the monthly PBL $\delta^{13}$C-$CO_2$ simulation results of





Chen et al. (2006) who showed that $\delta^{13}$C-CO$_2$ is 0.6‰ higher above the PBL than in the surface layer
near the ground. Recently, Ghasemifard et al. (2019) showed that hourly $\delta^{13}$C-CO$_2$ values at Mount
Zugspitze, the highest (2650 m) mountain in Germany, were about -7‰ in the winter for 2013. During an
especially clean air event (10 days in October) at Mount Zugspitze, the average $\delta^{13}$C-CO$_2$ was
approximately -7.5‰, which is consistent with our estimates using strategies 2 and 3. Based on the
evidence presented above, we believe that strategy 3 is the most robust way to derive a background $\delta^{13}$C-
CO$_2$ for the domain.
**3.2.2 Evaluation of $\delta^{13}$C-CO$_2$ simulations**
Figure 7a shows the hourly $\delta^{13}$C-CO$_2$ simulations over a two-year period. To the best of our knowledge,
this is the first time that $\delta^{13}$C-CO$_2$ has been simulated at an hourly time scale for an urban region. The
simulations are consistent with the observations at daily, monthly and annual time scales, where the
average value of observations (simulations) were -8.69‰ (-8.68‰) and -8.52‰ (-8.45‰) for 2014 and
2015, respectively. The corresponding correlation was R = 0.54 ($P < 0.001$) and R = 0.52 ($P < 0.001$).
The root mean square error between observations and simulations was 1.07‰ for 2014 and 1.10‰ for
2015 (Table 2). Further, the observed and simulated $\delta^{13}$C-CO$_2$ values showed seasonal variations that
increased in summer and decreased in winter. This pattern mirrored the CO$_2$ mixing ratios for both
observations and simulations (Figures 3 and 7). Similar relations and seasonal variations of $\delta^{13}$C-CO$_2$
have been reported in other urban areas (Sturm et al., 2006; Guha & Ghosh, 2010; Moore & Jacobson,
2015; Pang et al., 2016). The simulated hourly NEE CO$_2$ enhancement is also shown in Figure 7b. Note
that negative values indicate net CO$_2$ sinks and positive values indicate net CO$_2$ sources. We can see large
hourly variations in the growing seasons and positive enhancements during nighttime that are generally
larger than negative enhancements during daytime. This shows the potential influence of NEE on $\delta^{13}$C-
CO$_2$ seasonality. To date, no study has quantified the relative contributions to the $\delta^{13}$C-CO$_2$ seasonality.
Here, we re-evaluate and quantify the main factors contributing to its seasonality based on the
combination of $\delta^{13}$C-CO$_2$ observations and simulations in the following section.
Here, we examine the comparisons for winter and summer in greater detail. The simulations showed that
the model can generally capture the diel variations of observed hourly $\delta^{13}$C-CO$_2$ variations (Figure 8).
Statistics between observations and simulations for two seasons are shown in Table 2. The observed
seasonal average significantly increased, by 1.18‰, from winter 2013-2014 (-9.27‰) to summer 2014 (-
8.09‰). The simulations showed a similar seasonal increase of 1.35‰. Some large discrepancies are
evident and generally caused by the simulated total CO$_2$ enhancement biases and the negative relationship
between $\delta^{13}$C-CO$_2$ and the CO$_2$ enhancement.


Comparisons between observations and simulations for daily average $CO_2$ mixing ratio and $\boldsymbol{\delta}^{13}$C-$CO_2$ are
also shown in Figure 9. Although the data are distributed around the 1:1 line for both seasons, there is less
scatter and higher correlation in the winter than in the summer. We attributed this to the more complex
biological $CO_2$ sinks in the summer, which are not adequately resolved by the relatively coarse model
grid ($1^o$ by $1^o$).
**3.2.3 Mechanisms controlling the $\boldsymbol{\delta}^{13}$C-$CO_2$ seasonality**
The mechanisms driving these seasonal variations are examined below. The peak and trough in the
observed $\boldsymbol{\delta}^{13}$C-$CO_2$ signal was observed in December and July, respectively, yielding an amplitude of
1.51‰. This was consistent with the simulated amplitude of 1.53‰. These results support that the
simulated $\boldsymbol{\delta}^{13}$C-$CO_2$ seasonality agreed well with the observations (Figure 10), and can be used to further
diagnose the mechanisms contributing to the $\boldsymbol{\delta}^{13}$C-$CO_2$ seasonality. According to equation 2, the $\boldsymbol{\delta}^{13}$C-
$CO_2$ seasonality can be attributed to four factors including: (1) A change in the background $\boldsymbol{\delta}^{13}$C-$CO_2$
value from -7.64‰ in December to -6.66‰ in July; (2) A change in $CO_2$ background from 399 ppm to
398 ppm; (3) The total $CO_2$ enhancement change from 45.7 ppm to 37.3 ppm; and (4) The change in the
isotope composition of the $CO_2$ enhancements causing $\boldsymbol{\delta}$s to vary from -26.1‰ to -22.8‰.
To quantify each mechanism's contribution to the seasonality of atmospheric $\boldsymbol{\delta}^{13}$C-$CO_2$, we recalculated
$\boldsymbol{\delta}^{13}$C-$CO_2$ by using the monthly averages as described above. First, we calculated $\boldsymbol{\delta}^{13}$C-$CO_2$ in December
and July, which were -9.54‰ and -8.04‰, respectively, with amplitude of 1.50‰. Next, we replaced the
$\boldsymbol{\delta}^{13}$C-$CO_2$ background value in December (-7.64‰) with July (-6.67‰). The recalculated $\boldsymbol{\delta}^{13}$C-$CO_2$ was -
8.66‰ in December, indicating that the change in $\boldsymbol{\delta}^{13}$C-$CO_2$ background value caused a change of 0.88‰
(9.54‰ minus -8.66‰) to the seasonality. By changing both the total $CO_2$ enhancement and background
values, the recalculated $\boldsymbol{\delta}^{13}$C-$CO_2$ was -8.32‰, contributing a 0.34‰ change in the seasonality (-8.66‰
minus -8.32‰). Finally, by changing $\boldsymbol{\delta}$s from -26.1‰ to -22.8‰, together with the change in background
value, the recalculated $\boldsymbol{\delta}^{13}$C-$CO_2$ was -8.32‰ –a change of 0.34‰ (i.e. -8.66‰ minus -8.32‰). This
indicates that both the total $CO_2$ enhancement and change in $\boldsymbol{\delta}$s contributed equally to the regional source
term, causing a variation of 0.62‰ (i.e. 1.50‰ minus 0.88‰). Based on the above analyses, we attributed
59% and 41% of the $\boldsymbol{\delta}^{13}$C-$CO_2$ seasonality to the changing $\boldsymbol{\delta}^{13}$C background term and regional source
terms, respectively. Further, the total $CO_2$ enhancement and $CO_2$ enhancement components contributed
equally (about 20%) to the $\boldsymbol{\delta}^{13}$C-$CO_2$ seasonality.
To investigate how ecosystem photosynthetic discrimination and respiration affected atmospheric $\boldsymbol{\delta}^{13}$C-
$CO_2$ seasonality, we simulated the $\boldsymbol{\delta}^{13}$C-$CO_2$ again for two cases: (1) excluding photosynthetic
discrimination, and (2) excluding both photosynthetic discrimination and respiration. Note that only NEE



was used in our study with no partitioning between photosynthesis and respiration in the daytime.
Hereafter, we use negative NEE to define case 1 when photosynthesis exceeded respiration. The results
are shown in Figure 10 b-c. Overall, the negative $CO_2$ enhancement (i.e. photosynthesis > respiration)
caused atmospheric $\boldsymbol{\delta}^{13}C$-$CO_2$ to become more enriched than the baseline simulations with maximum
values around 1‰ between April and October (Figure 10b), and positive $CO_2$ enhancement (i.e. via
respiration) caused atmospheric $\boldsymbol{\delta}^{13}C$-$CO_2$ to become more depleted compared to the baseline simulations
through the whole year (Figure 10c). By applying the CCGRCV fitting technique to the $\boldsymbol{\delta}^{13}C$-$CO_2$ for the
above two cases, we found that the $\boldsymbol{\delta}^{13}C$-$CO_2$ seasonality decreased to 1.45‰ in case 1, indicating
ecosystem photosynthetic discrimination explained only 0.08‰ of the seasonality (1.53‰ minus 1.45‰).
For case 2, the $\boldsymbol{\delta}^{13}C$-$CO_2$ trough in winter slightly increased by 0.08‰ and peak in summer increased by
0.20‰, these two factors finally lead the seasonality increase to 1.66‰, which were caused by much
larger respiration $CO_2$ enhancement in summer than in winter (Figure 7b). These results indicate that
biological respiration reduced the $\boldsymbol{\delta}^{13}C$-$CO_2$ seasonality by 0.20‰, and that negative NEE (photosynthetic
discrimination) acted to increase the $\boldsymbol{\delta}^{13}C$-$CO_2$ seasonality by 0.08‰. Generally, ecosystem
photosynthesis played a minor role in controlling the atmospheric $\boldsymbol{\delta}^{13}C$-$CO_2$ seasonality within this urban
area. In other words, the anthropogenic $CO_2$ emissions played a much larger role than the plants.
As shown in Figure 5, $CO_2$ sources from power industry, combustion for manufacturing, non-metallic
mineral production and oil refineries and transformation industry were the top 4 contributors to the $CO_2$
enhancements. We simulated atmospheric $\boldsymbol{\delta}^{13}C$-$CO_2$ by assuming that no $CO_2$ was emitted from each of
these 4 categories. The simulations were performed by excluding one category at a time. The results
indicated that atmospheric $\boldsymbol{\delta}^{13}C$-$CO_2$ seasonality was 1.30‰, 1.57‰, 1.30‰, and 1.47‰, if excluding
power industry, combustion for manufacturing source, oil refineries/transformation industry, and non-
metallic mineral production sources, respectively. In other words, power industry and oil refineries/
transformation industry together contributed a 0.40‰ to the total regional source term of 0.62‰. The
cement sources played a role in enriching (0.05‰ to 0.07‰) the atmospheric $\boldsymbol{\delta}^{13}C$-$CO_2$ in the heavy
isotope, contrary to all other anthropogenic $CO_2$ sources.
**3.3 Sensitivity analysis**
**3.3.1 Comparison of $\boldsymbol{\delta}s \cdot \Delta CO_2$**
Based on equation 2, the regional source term determines the hourly/daily variations of $\boldsymbol{\delta}^{13}C$-$CO_2$, which
is treated as a signal added to the background signal. To evaluate the model simulated regional source
term with respect to the observations we examined daily averages for winter to minimize the influence of
photosynthesis. In Figure 11a, the observed daily $\boldsymbol{\delta}s \cdot \Delta CO_2$ values are compared with the simulated values





using the *a priori* anthropogenic $CO_2$ emissions. Here $\Delta CO_2$ represents the total $CO_2$ enhancement for
both observations and simulations. The product $\delta s \cdot \Delta CO_2$ can be interpreted as the regional source term.
The average values were -1009.0 (and -841.9) ppm·‰ for observations and -1096.7 (and 1000.5) ppm·‰
for model results in 2014 (and 2015). The slope of the regression fit was 0.99 (±0.12) and the intercept
was -151.7 (±130.1) for all data during the two winters. After applying the monthly scaling factors to
constrain the anthropogenic $CO_2$ emissions, the re-calculated results were closer to the 1:1 line with a
slightly improved correlation (R increased from 0.47 to 0.50; Figure 11b). Note that the application of the
monthly scaling factors only impacts the $\Delta CO_2$ but not $\delta s$. The uncertainty in $\delta s$ will be discussed next.
**3.3.2 Comparison between $\delta_{ms}$ and $\delta_s$**
To evaluate the $\delta_s$ simulations, we compared observed and simulated $\delta_s$ as displayed in Figure 12a for all-
day and nighttime conditions. Here, nighttime simulations were selected to minimize the effects of
ecosystem photosynthesis and to focus on the anthropogenic $CO_2$ sources. Two methods were used to
calculate $\delta_s$ from the observations including the Miller-Tans and Keeling plot methods. Although $\delta_s$
differed between these two methods, both displayed similar seasonal variations with higher values ($\delta^{13}$C
enrichment) in summer and lower values in winter.  Such seasonal variations were also observed at other
urban sites including Beijing, China (Pang et al., 2016), Bern, Switzerland (Sturm et al., 2006), Bangalore
city, India (Guha and Ghosh, 2010),Wroclaw, Poland (Górka and Lewicka-szczebak, 2013).
If the $CO_2$ sources/sinks are homogeneously distributed and without monthly variations, the atmospheric
$CO_2$ enhancement components would remain unchanged, and there would be no seasonal changes in $\delta_s$.
In reality, variations in atmospheric transport processes interact with regional $CO_2$ sink/source changes
that cause monthly variations in $\delta_s$. The comparison of $\delta_s$ between simulations and observations indicated
that the model performed well in capturing the mixing and transport of $CO_2$ from different sources. We
can also infer from their difference that the proportions of some $CO_2$ categories were biased in the *a*
*priori* emission map. This can be caused by both the downscaling of EDGAR inventory distribution to
0.1° and the magnitude of some emissions categories. Among all anthropogenic sources, the most
significant linear relations were found between the simulated anthropogenic $\delta_s$ and cement $CO_2$
proportions for these 24 months, with slopes of 0.33‰ for nighttime and 0.35‰ for all-day conditions ($R^2$
= 0.97, p < 0.001; Figure 12 b & c). These results strongly support our hypothesis that cement $CO_2$
emissions dominated monthly $\delta s$ variations in the YRD region.
**3.3.3 Sensitivity of atmospheric $\delta^{13}$C-$CO_2$ and $\delta_s$ to cement $CO_2$ emissions**





The discrepancy between simulated and observed $\delta_s$ highlights that some $CO_2$ sources were biased in the
*a priori* inventories. As discussed above, cement $CO_2$ emissions had the most distinct $\delta^{13}C\text{-}CO_2$ end-
member value of 0‰ ± 0.30‰. Combined with its large emission, it had a strong potential to influence $\delta_s$
and $\delta^{13}C\text{-}CO_2$. YRD represents the largest cement producing region in the world. Its relative proportion
to total national anthropogenic $CO_2$ emissions is about 5.5% to 6.5% based on IPCC method and 7.3% for
EDGAR. These proportions are 50% greater than the global average of 4% (Boden et al., 2016) and much
larger than most countries (Andrew, 2018) and other large urbanized areas such as California (2%; Cui et
al., 2019).
The local activity data reveals that the cement production increased from $3.55 \times 10^8$ tons in 2010 to $4.56 \times$
$10^8$ tons in 2014 in the YRD area. Our own calculation of the national clinker-to-cement indicated a
decreasing trend from 64% in 2004 to around 56% in 2015. Here, we applied the value of 61.7% for 2010
and the average value of 57.0% for 2014 to 2015. We then used the EF for clinker (0.52 ± 0.01 tonne $CO_2$
per tonne clinker; IPCC 2013). Finally, the calculated cement $CO_2$ emissions were 1.14 (± 0.02) $\times 10^8$
tonne for 2010 and 1.35 ($\pm0.03$) $\times 10^8$ tonne for 2014, indicating an 18.4% increase over this time period.
This result is close to the scaling factor 1.145 for the total anthropogenic $CO_2$ emissions for the same
period.
The cement $CO_2$ emission was $1.45\times10^8$ tonne for the EDGAR products in 2010. Applying the scaling
factor of 1.184, based on our independent method, the EDGAR cement $CO_2$ emissions was $1.72\times10^8$
tonne for the year of 2014. The 27% difference between the EDGAR inventory and our independent
calculations probably resulted from large errors in the clinker-to-cement ratio and regional activity data.
Ke et al. (2013) reported a much higher clinker-to-cement ratio of 73% to 70% for China during 2005 and
2007 than the ratio of 57% in 2014 to 2015. If we applied a 70% ratio, the EDGAR cement $CO_2$ emission
would change to $1.28\times10^8$ tonne for 2010.
The monthly cement emission proportions varied from 6.21% to 8.98%, while its enhancement proportion
was much larger and could reach 16.85%. In other words, favorable atmospheric transport processes
amplified the cement $CO_2$ enhancement proportion at our observational site (Table S2). To quantify the
extent to which the cement $CO_2$ enhancement components can affect $\delta_s$ and atmospheric $\delta^{13}C\text{-}CO_2$ we
conducted sensitivity tests by changing the cement enhancement proportions to 0.8, 1.2, 1.4, 1.6, 1.8, and
2 times its original value. These sensitivity tests are based on two different assumptions for cement $CO_2$
enhancement changes: (1) There is no bias in the total anthropogenic $CO_2$ enhancement such that a
proportional increase/decrease in the cement component does not change the relative anthropogenic
contributions; (2) Only the cement enhancement changes. From equation 2, these two assumptions will
change both $\delta_s$ and $\delta^{13}C\text{-}CO_2$ but with different amplitude.



Results for the first assumption are shown in Figure 13a-b for both nighttime and all-day $\delta_s$ simulations.
The simulated $\delta_s$ increased linearly with the increase of cement proportions, at a rate of 2.73‰ increase
per 10% increase of cement proportions in the nighttime and 2.72‰ for all-day. The result for the second
assumption is relatively similar with the first one, yielding a 2.32‰ increase for a 10% increase in the
cement proportion. As shown in Table S2, the cement $CO_2$ enhancement proportions increased from 5.60%
- 6.77% (December) to 13.16% - 16.85% (June), which is the primary cause for the observed monthly $\delta_s$
variations. The high sensitivity of $\delta_s$ to cement $CO_2$ proportions can partly explain the relative difference
of modeled $\delta_s$ and indicates a potential advantage to constrain cement $CO_2$ emissions by using
atmospheric $\delta^{13}$C-$CO_2$ observations. Finally we calculated how cement $CO_2$ can change atmospheric
$\delta^{13}$C-$CO_2$ (Figure 13c). These results show that atmospheric $\delta^{13}$C-$CO_2$ is more sensitive to the first
assumption than the second assumption. These sensitivity analyses indicate that a cement $CO_2$
enhancement relative change of 20% (or 1.57% increase) can cause a 0.013‰ - 0.038‰ change in the
atmospheric $\delta^{13}$C-$CO_2$. These results indicate that $\delta_s$ is more sensitive to cement $CO_2$ emissions compared
with other anthropogenic and biological $CO_2$ sources/sinks.
**4 Conclusions**
(1) Total annual anthropogenic $CO_2$ emissions for the YRD showed high consistency between the top-
down and bottom-up approaches with a bias less than 6%.
(2) Approximately 59% and 41% of the $\delta^{13}$C-$CO_2$ seasonality were attributed to the change in $\delta^{13}$C
background value and the regional $CO_2$ source term, respectively.
(3) Power industry and oil refineries/ transformation industry together contributed 0.40‰, accounting
for 64.5% of all regional source terms (0.62‰).
(4) If excluding all ecosystem respiration and photosynthetic discrimination in YRD area, $\delta^{13}$C-$CO_2$
seasonality will increase from 1.53‰ to 1.66‰.
(5) Atmospheric transport processes in summer amplified the cement $CO_2$ enhancement proportions in
the YRD area, which dominated monthly $\delta$s variations. $\delta_s$ was shown to be a strong linear relation
with cement $CO_2$ proportion in the YRD area.
**Acknowledgements**
This research was partially supported by start-up foundation (163108094) from Nanjing Forestry
University, Natural Science Foundation of Jiangsu Province (BK20181100), and
Key Research Foundation of Jiangsu Meteorological Society (KZ201803).
**Code/Data availability**



The data presented in this manuscript has been uploaded on our group website:
https://yncenter.sites.yale.edu/data-access.
**Author contribution:** Cheng Hu, Timothy J. Griffis and Xuhui Lee designed the study, Cheng
Hu performed the model simulation, Cheng Hu write the original draft, Supervision: Timothy J.
Griffis and Xuhui Lee, Data acquisition: Jiaping Xu, Wenjing Huang, Dong Yang, Yan Chen,
Cheng Liu, Shoudong Liu, and Lichen Deng,  all co-authors contributed to the data analysis.
**Competing interests**: The authors declare that they have no conflict of interest.

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



**Figure 1**. (a) Weather Research and Forecasting Model simulation domains and the location of WLG site, (b) cement production distribution in YRD and Eastern China.

**Figure 2**. (a) Annual anthropogenic $CO_2$ emissions for study domain (units: nmol m$^{-2}$ s$^{-1}$) and population density in 4 megacities (units: people per hectare) including Nanjing, Hefei, Zhejiang, and Shanghai for the year of 2015, (b) Two-year average concentration footprint.

**Figure 3**. (a) Comparisons of hourly $CO_2$ mixing ratios between observations and model simulation from September 2013 to August 2015, and monthly averages for (b) whole day, (c) nighttime (22:00-06:00, local time) and (d) daytime (10:00 - 16:00); Model results (red), observations (black), and background (grey).

**Figure 4**. (a) Relation between monthly PBL height and change in $CO_2$ mixing ratio; Time series (2013 to 2015) of (b) NDVI, (c) SIF, and (d) GPP.

**Figure 5**. (a) Comparisons of simulated and observed $CO_2$ enhancement, (b) Simulated anthropogenic $CO_2$ enhancement proportion for the main sources, and (c) $CO_2$ enhancement contributions from different provinces.

**Figure 6**. Comparisons among three strategies for calculating the background $\boldsymbol{\delta}^{13}$C-$CO_2$ . Strategy 1 (WLG discrete: weekly discrete observations at WLG site, WLG CCGCRV: derived hourly data with WLG observations and CCGCRV method); Strategy 2 (Calculated: by choosing clean air in winter); and strategy 3 (M-T method: derived results with observations and M-T approach, M-T CCGCRV: derived hourly results with M-T approach and CCGCRV method, see details in section 2.2.1).

**Figure 7**. (a) Comparisons of observed and modeled hourly $\boldsymbol{\delta}^{13}$C-$CO_2$ from September 2013 to August 2015, and (b) Simulated hourly biological $CO_2$ enhancement.

**Figure 8**. Comparisons of observed and modeled (a) $CO_2$ mixing ratio and (b) $\boldsymbol{\delta}^{13}$C-$CO_2$ from December 2013 to February 2014; (c) $CO_2$ mixing ratio and (b) $\boldsymbol{\delta}^{13}$C-$CO_2$ from December 2014 to February 2015; (e) $CO_2$ mixing ratio and (f) $\boldsymbol{\delta}^{13}$C-$CO_2$ from June 2014 to August 2014; (g) $CO_2$ mixing ratio and (h) $\boldsymbol{\delta}^{13}$C-$CO_2$ from June 2015 to August 2015.

**Figure 9**. Scatter plots of observed versus modeled (a) winter time $CO_2$ mixing ratios, (b) winter time $\boldsymbol{\delta}^{13}$C-$CO_2$, (c) summer time $CO_2$, and (d) summer time $\boldsymbol{\delta}^{13}$C-$CO_2$ for both years.

**Figure 10**. Digital filtering curve fitting (CCGCRV) for background, observations, normal simulations, case 1 (excluding photosynthesis), and case 2 (excluding respiration and photosynthesis) in both years, (b) $\boldsymbol{\delta}^{13}$C-$CO_2$ comparisons between normal simulations and case 1, and (c) $\boldsymbol{\delta}^{13}$C-$CO_2$ comparisons between normal simulations and case 2.

**Figure 11**. Comparisons of winter time $\boldsymbol{\delta}$s·$\boldsymbol{\Delta}CO_2$ using (a) *a priori* and (b) constrained anthropogenic $CO_2$ emissions.

**Figure 12**. (a) Comparisons between observed and modeled $\boldsymbol{\delta}_s$, (b) relationship between cement $CO_2$ enhancement proportion and simulated anthropogenic $\boldsymbol{\delta}$s for nighttime and (c) all-day.

**Figure 13**. Sensitivity tests showing the influence of cement $CO_2$ emissions on $\boldsymbol{\delta}_s$ for (a) nighttime, (b) all-day, and (c) the relation between cement $CO_2$ and $\boldsymbol{\delta}^{13}$C for simulation strategies 1 and 2. Note that the numbers in brackets indicate changes in $\boldsymbol{\delta}^{13}$C with cement $CO_2$ proportion increase by 0.2 times. The x-axis values indicate changing cement enhancement proportions to 0.8 1.2, 1.4, 1.6, 1.8, and 2 times the original values.



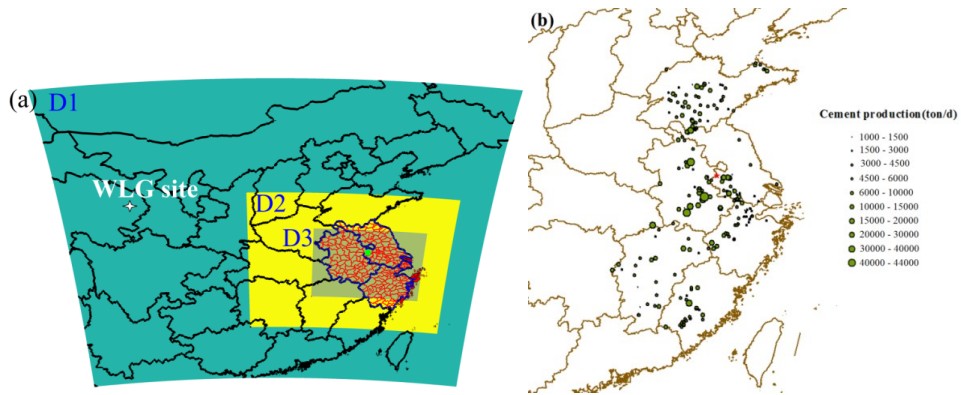



Figure 1. (a) Weather Research and Forecasting Model simulation domains and the location of WLG site, (b) cement production
distribution in YRD and Eastern China.

























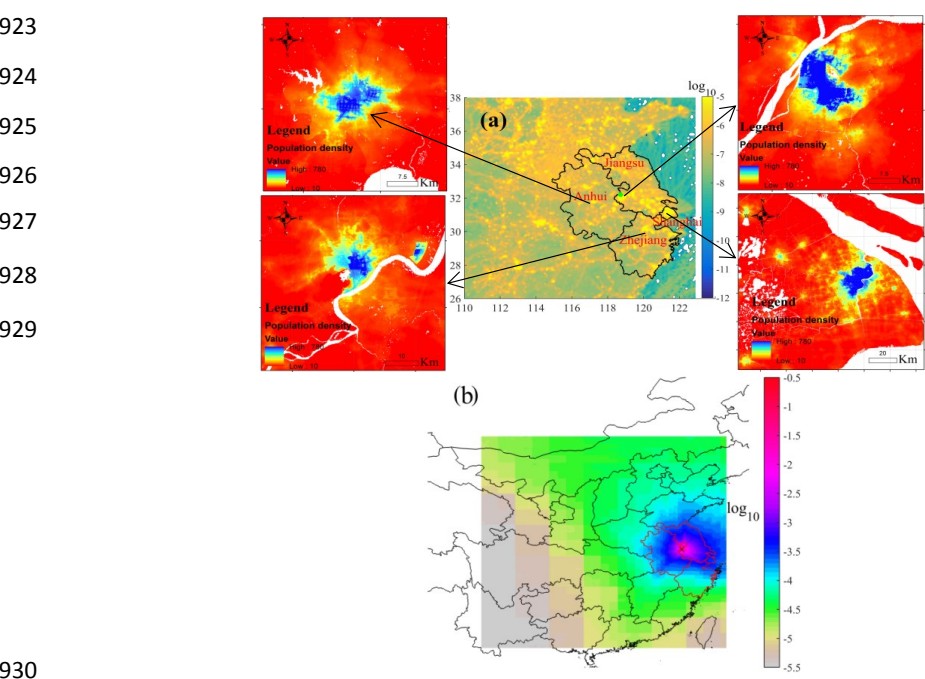


Figure 2. (a) Annual anthropogenic $CO_2$ emissions for study domain (units: nmol m$^{-2}$ s$^{-1}$) and population density in 4 megacities
(units: people per hectare) including Nanjing, Hefei, Zhejiang, and Shanghai for the year of 2015, (b) Two-year average
concentration footprint.







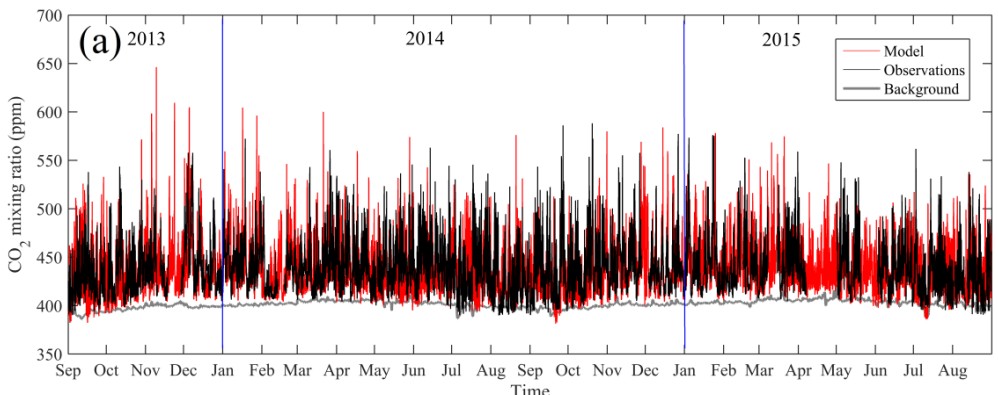


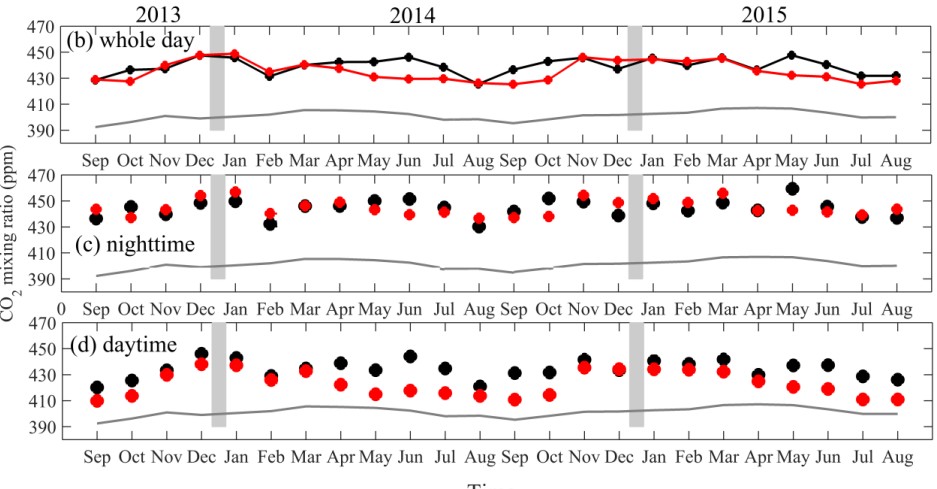


Figure 3. (a) Comparisons of hourly $CO_2$ mixing ratios between observations and model simulation from September 2013 to August 2015, and monthly averages for (b) whole day, (c) nighttime (22:00-06:00, local time) and (d) daytime (10:00 - 16:00); Model results (red), observations (black), and background (grey).







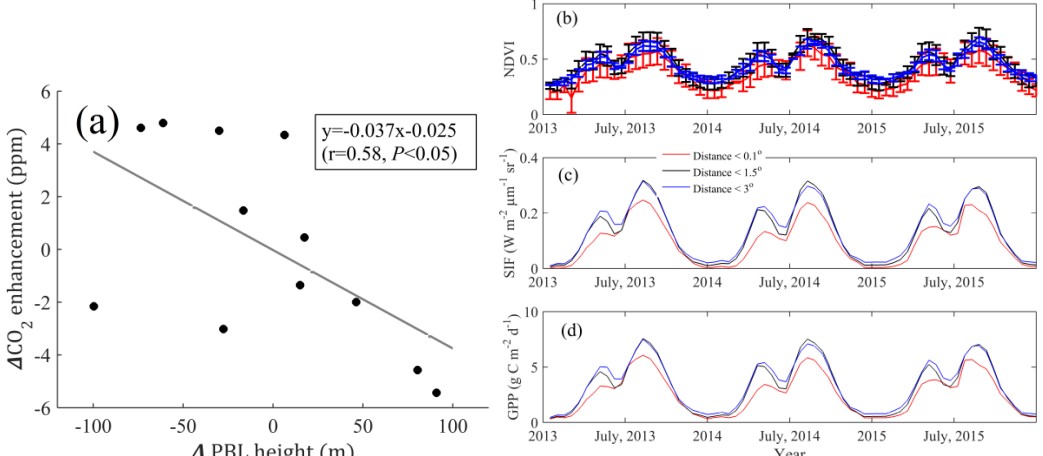



Figure 4. (a) Relation between monthly PBL height and change in $CO_2$ mixing ratio; Time series (2013 to 2015) of (b) NDVI, (c)
SIF, and (d) GPP.










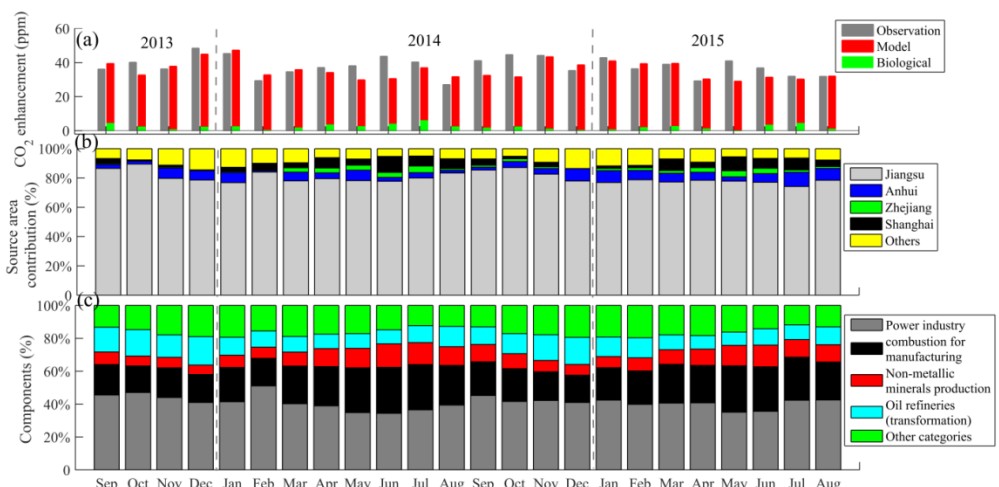


Figure 5. (a) Comparisons of simulated and observed $CO_2$ enhancement, (b) Simulated anthropogenic $CO_2$ enhancement
proportion for the main sources, and (c) $CO_2$ enhancement contributions from different provinces.



















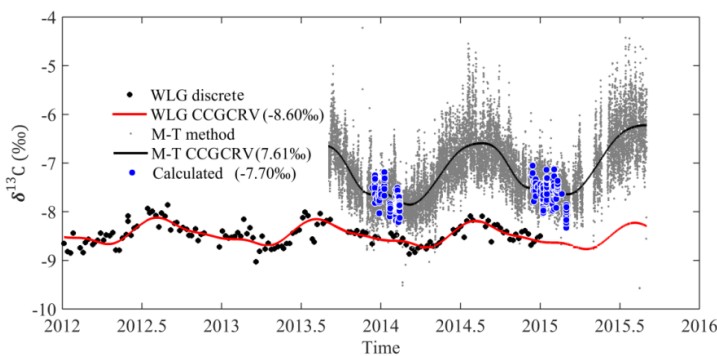


Figure 6. Comparisons among three strategies for calculating the background $\delta^{13}$C-CO$_2$ . Strategy 1 (WLG discrete: weekly

discrete observations at WLG site, WLG CCGCRV: derived hourly data with WLG observations and CCGCRV method);

Strategy 2 (Calculated: by choosing clean air in winter); and strategy 3 (M-T method: derived results with observations and M-T

approach, M-T CCGCRV: derived hourly results with M-T approach and CCGCRV method, see details in section 2.2.1).





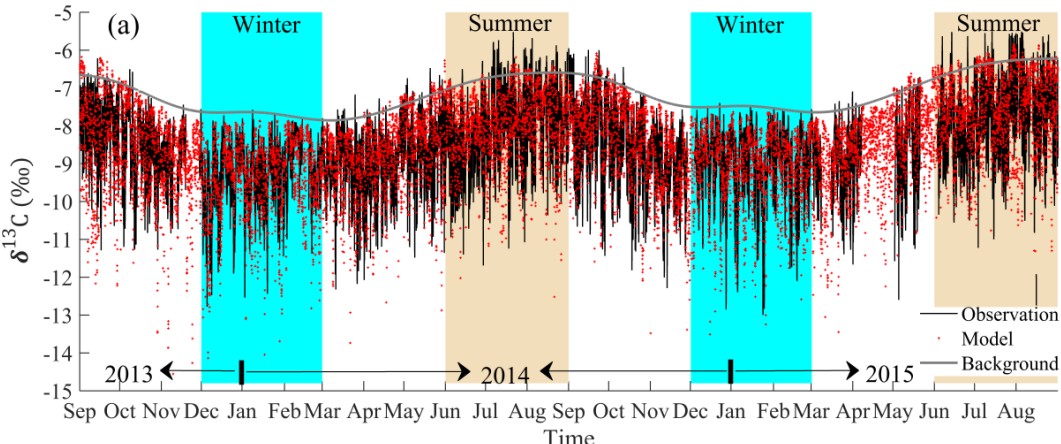



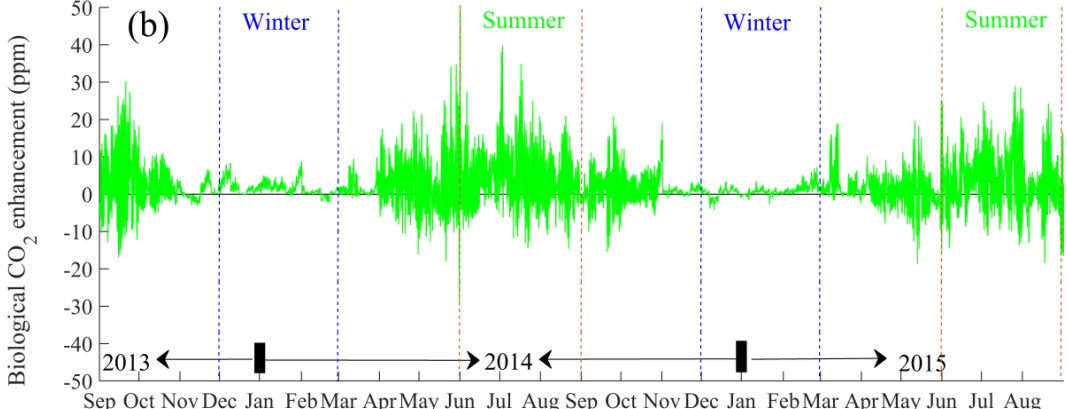


Figure 7. (a) Comparisons of observed and modeled hourly $\delta^{13}$C-CO$_2$ from September 2013 to August 2015, and (b) Simulated
hourly biological CO$_2$ enhancement.




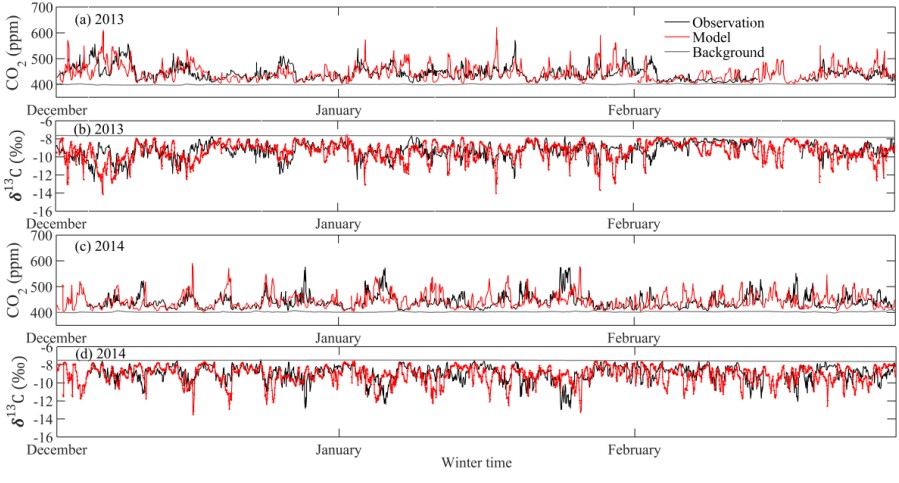


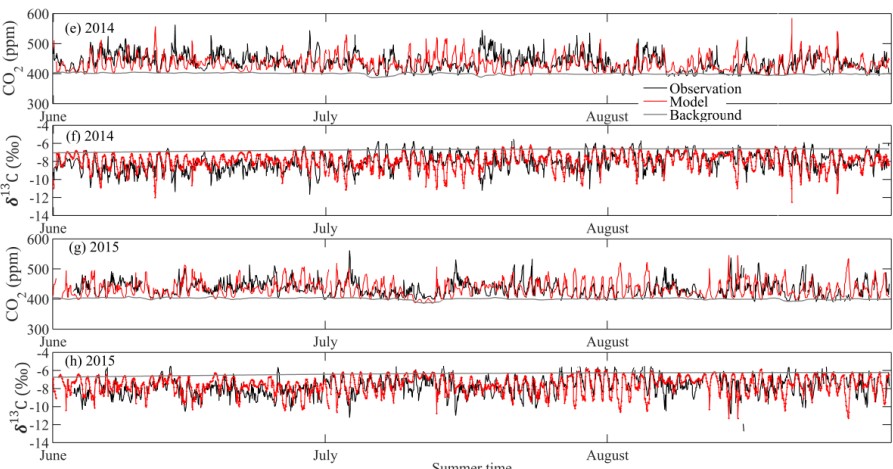


Figure 8. Comparisons of observed and modeled (a) $CO_2$ mixing ratio and (b) $\delta^{13}C$-$CO_2$ from December 2013 to February 2014;

(c) $CO_2$ mixing ratio and (b) $\delta^{13}C$-$CO_2$ from December 2014 to February 2015; (e) $CO_2$ mixing ratio and (f) $\delta^{13}C$-$CO_2$ from

June 2014 to August 2014; (g) $CO_2$ mixing ratio and (h) $\delta^{13}C$-$CO_2$ from June 2015 to August 2015.





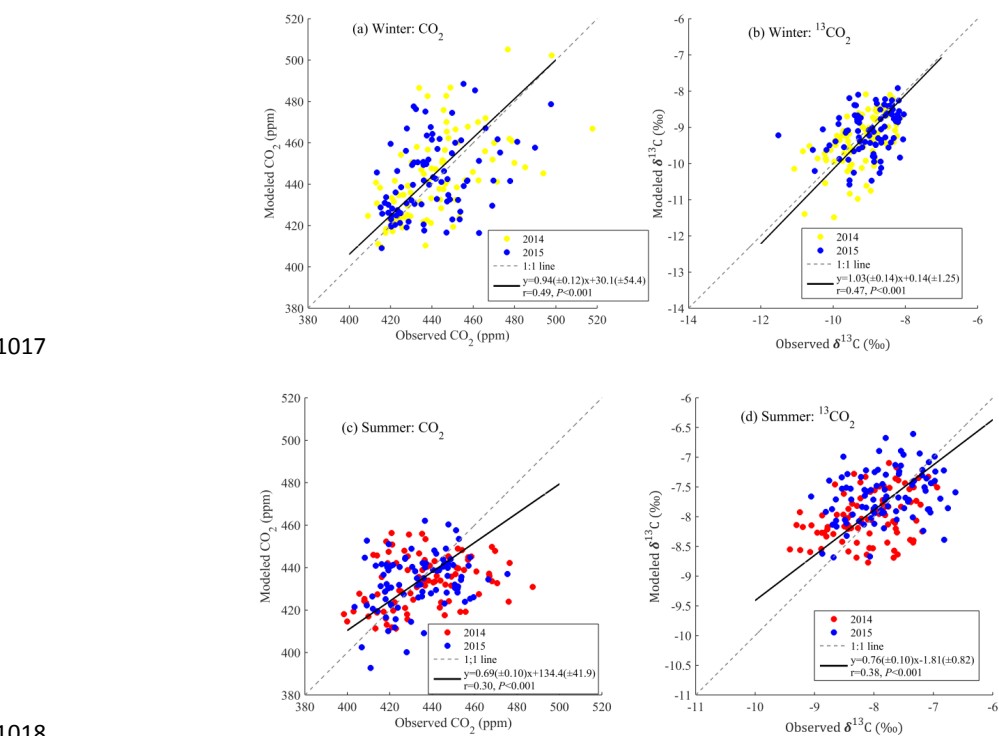


Figure 9. Scatter plots of observed versus modeled (a) winter time $CO_2$ mixing ratios, (b) winter time $\delta^{13}C$-$CO_2$, (c) summer time
$CO_2$, and (d) summer time $\delta^{13}C$-$CO_2$ for both years.











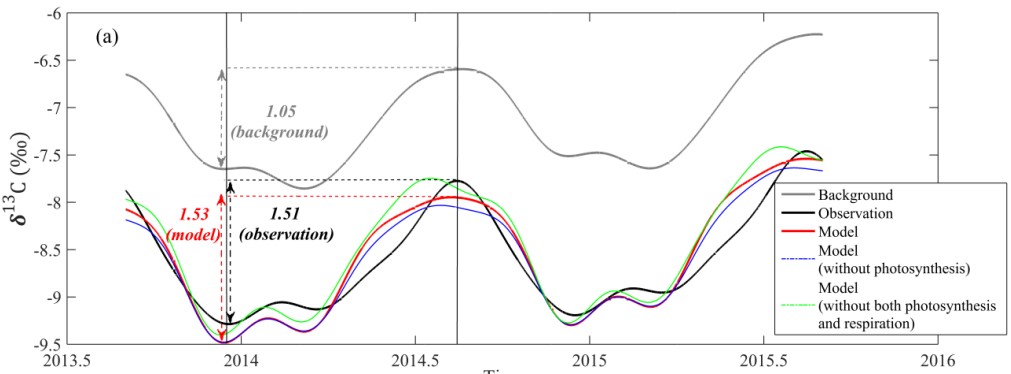


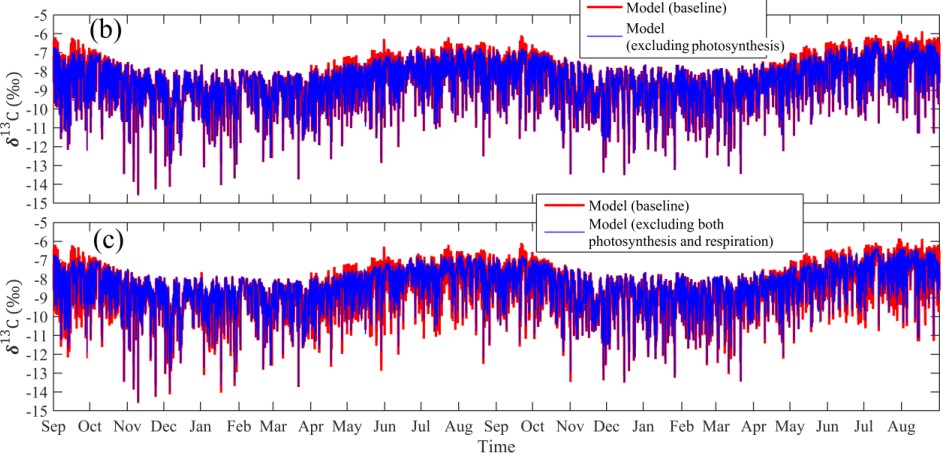

Figure 10. Digital filtering curve fitting (CCGCRV) for background, observations, normal  simulations, case 1 (excluding
photosynthesis), and case 2 (excluding respiration and photosynthesis) in both years, (b) $\delta^{13}$C-CO$_2$ comparisons between normal
simulations and case 1, and (c) $\delta^{13}$C-CO$_2$ comparisons between normal simulations and case 2.






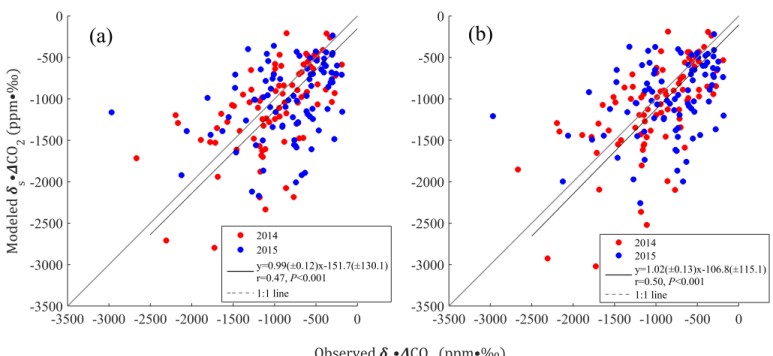


Figure 11. Comparisons of winter time $\delta$s·$\Delta$CO$_2$ using (a) *a priori* and (b) constrained anthropogenic CO$_2$ emissions.


















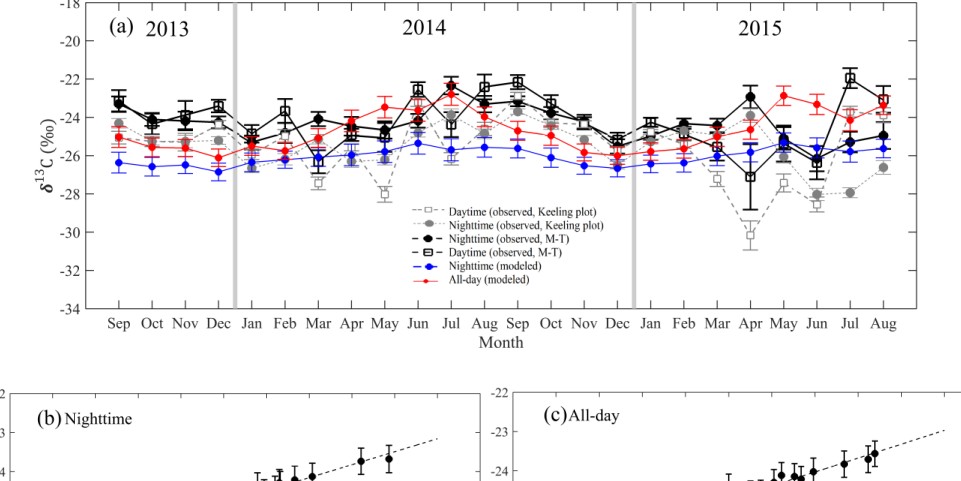


Figure 12. (a) Comparisons between observed and modeled $\delta_s$, (b) relationship between cement $CO_2$ enhancement proportion and
simulated anthropogenic $\delta$s for nighttime and (c) all-day.













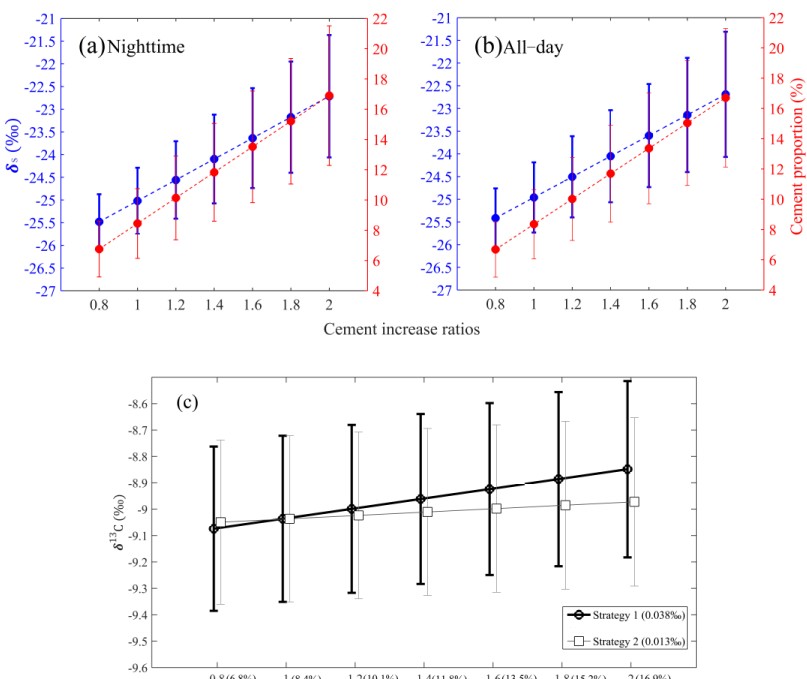



Figure 13. Sensitivity tests showing the influence of cement $CO_2$ emissions on $\delta_s$ for (a) nighttime, (b) all-day, and (c) the relation between cement $CO_2$ and $\delta^{13}C$ for simulation strategies 1 and 2. Note that the numbers in brackets indicate changes in $\delta^{13}C$ with cement $CO_2$ proportion increase by 0.2 times. The x-axis values indicate changing cement enhancement proportions to 0.8 1.2, 1.4, 1.6, 1.8, and 2 times the original values.


















1080         **Table 1.** Comparisons of cement and all anthropogenic $CO_2$ emissions among different methods.

| Units: $\times 10^{11}$ kg | Year | EDGAR v432 | Inversion results | IPCC method |
|---|---|---|---|---|
| Cement $CO_2$ emissions | 2010 | 1.45 | / | 1.14 |
| | 2014-2015 | 1.72 | / | 1.35 |
| All anthropogenic $CO_2$ emissions | 2010 | 20.55 | / | 17.56 |
| | 2014-2015 | 23.53 | $24.59 \pm 2.39$ | 24.38 |






















**Table 2**. Statistical metrics between observed and modeled $CO_2$ mixing ratios and $\boldsymbol{\delta}^{13}$C-$CO_2$ during winter, summer and annual
for 2014 and 2015. Correlation coefficient (R), mean bias (MB), and root mean square error (RMSE) are displayed.

| | Years | 2014 | | | 2015 | | |
|---|---|---|---|---|---|---|---|
| | Periods | Annual | Winter | Summer | Annual | Winter | Summer |
| $\boldsymbol{\delta}^{13}$C-$CO_2$ | R | 0.54 | 0.40 | 0.47 | 0.52 | 0.27 | 0.39 |
| | RMSE (‰) | 1.07 | 0.94 | 0.94 | 1.10 | 0.92 | 0.98 |
| | simulation (‰) | -8.68 | -9.37 | -8.02 | -8.45 | -9.10 | -7.66 |
| | observation (‰) | -8.69 | -9.27 | -8.09 | -8.52 | -8.98 | -7.83 |
| $CO_2$ | R | 0.38 | 0.41 | 0.34 | 0.35 | 0.28 | 0.31 |
| | RMSE (ppm) | 29.44 | 27.48 | 25.55 | 30.22 | 26.81 | 24.29 |
| | MB (ppm) | 2.16 | -0.27 | 3.80 | 2.99 | -0.43 | 1.53 |

