# Peer review of "Anthropogenic and natural controls on atmospheric δ13C-CO2 variations in the Yangtze River Delta: Insights from a carbon isotope modeling framework"

_Atmospheric Chemistry and Physics, 2020_

## Referee Comment (RC1) · Anonymous Referee #1 · 15 Oct 2020

Review of: Anthropogenic and natural controls on atmospheric $\delta^{13}$C-CO$_2$ variations in the Yangtze River Delta: Insights from a carbon isotope modeling framework

Authors: Cheng Hu, Jiaping Xu, Cheng Liu, Yan Chen, Dong Yang , Wenjing Huang, Lichen, Deng, Shoudong Liu, Timothy Griffis, and Xuhui Lee

This paper describes a study of CO$_2$ emissions in the megacity region of the Yangtze River Delta of China, which include several major cities in eastern China.  The novel contribution of this study is the WRF-STILT modeling of the emissions making extensive use of the stable isotopic composition of carbon in CO$_2$ ($\delta^{13}$C-CO$_2$).  The simulation agrees well with the CO$_2$ observations. The modeling of $\delta^{13}$C-CO$_2$ allows investigation of the contributions of various anthropogenic and biogenic sources.  The topic of this study falls well within the scope of Atmospheric Chemistry and Physics.  Therefore, this paper should be published after minor revision.

My concerns include the need for clarification and further discussion of several points and the need for quantification of uncertainties in calculations resulting from the modeling runs. Particular instances of these are given below.

Specific comments:
Line 114: Replace "be used" with "been used."
Line 188ff: Move Figure 6 here, since you are describing it here.  You should refer to it here, changing the number to 2, and therefore adjusting the figure numbers for the old 2-5 to 3-6, both in the text and in the figure captions.
Lines 205-206: The lowest quintile is the lowest 20%, not the lowest 5%.  You can just say "the bottom 5%" to describe the data used in this approach to background.
Lines 216-218: Give the parameters you used in the CCGCRV curve fitting calculations.
Line 231: Add (Figure 2 (perhaps changed to Figure 3)) after "YRD."
Line 243: Replace "East China" with "Eastern China."
Line 247: Insert "backwards" after "locations."
Line 250: Replace "at the end of" with "for."
Lines 263-264 and elsewhere: Replace "EDGAR v432" with "EDGAR v4.3.2."
Line 276: Please clarify "enhancement."  Is this proportion of emissions due to source $i$? "Enhancement" sounds like it is the amount of CO$_2$ in excess background.
Line 286: When considering the biosphere in cities, people are starting to include the effects of human respiration and excretion (Turnbull et al., 2015; Miller et al., 2020, just published in PNAS), using information from Prairie and Duarte (2007).  You might want to comment on how this would affect your analysis.  The $\delta^{13}$C of human respiration should reflect that of the average diet.  Also, is any bioethanol used in the gasoline?  You should confirm this, since this is common is some cities.
Line 322: Replace "blue" with "blue-red."
Lines 333-334: Replace "below" with "to."
Line 335: What hours did you use for daytime? Most modelers stress that mid-day to mid-afternoon hours work best, when the planetary boundary layer height (PBLH) is best simulated.

Your Figure 3 suggests that the monthly average of nighttime modeling results matches the observation best.

Line 345: What are the two months that fall far below the trend in Figure 4a? Do you have an explanation for these?

Line 359: Neither Figure 5a nor 7b is consistent with a negative average summer NEE. Indeed, Figure 5a suggests the opposite since all 2014 summer months are positive in NEE/biological contribution to the $CO_2$ enhancement, as are June and July 2015.

Line 363: Replace "4b-c" with "4b-d."

Lines 375-376: What are the uncertainties in the observed anthropogenic $CO_2$ enhancements? In general, please give uncertainties.

Lines 388-390: The absolute enhancements depend on many things, including the meteorology and the magnitude of the emissions. You can't conclude that the YRD has more emissions simply because the enhancements are higher. Modeling is critical for coming to that conclusion.

Line 392: Explicitly explain where these percentages come from.

Line 395: Where do you show that the maximum source contribution exceeded 50% on 19 September 2013?

Lines 396-399: Please explain how the "anthropogenic enhancement" is different from the "anthropogenic emissions."

Line 408: Replace "2014" with "2014-2015."

Lines 408 and 415: Express the emissions as $*10^{11}$ kg, the same units as in Table 1, for consistency.

Line 417: Replace "is" with "are."

Lines 430-431: A positive biological $CO_2$ signal during winter is consistent with a negligible role for photosynthesis, but it could be that photosynthesis is still important, just not as important as respiration. Will human respiration affect this?

Line 434: Replace "domain" with "background."

Line 435: Add "(Figure 6)" after "respectively." This may become Figure 2.

Line 454: Replace "Figures 3 and 7" with "Figures 3a and 8", but the vertical scale in Figure 8 is too compressed to be seen clearly.

Lines 465-469: Please give uncertainties. Are the seasonal increases significant? Could PBLH simulation issue explain the large discrepancies, especially since the model diurnal variations are greater than those in the observations.

Line 505: Replace "than" with "in." The baseline simulation in Figure 10b (red) is more enriched in the heavy isotope, as evidenced by its less negative values between April and October.

Lines 509ff: Are the differences of 0.08-0.20‰ significant? Please give uncertainties. Similar comment for the next paragraph.

Line 530: Add the definition of the regional source term "$(\delta_s * \Delta CO_2)$." In general, be consistent with "$\delta_s$"

Line 544: If you use nighttime simulations, you still have respiration.

Lines 597-598: Do you need to show both 13a and b? They are almost identical. You could just show one and generally state the results for the second in the text.

Lines 600-601: Replace "relatively similar with" with "similar to."

Lines 608-609: Insert "absolute" after "1.57%."  Are the uncertainties in your calculations small enough that "a 0.013‰ – 0.038‰ change" is significant?

Lines 621-622: Add "calculated from the simulations" before "was shown" and "From the EDGAR v4.3.2 inventory" after "proportion."

Figure 1: More information is needed in the caption – significance of the different-colored boundaries.  Is the red triangle in (b) the same as the blue dot in (a) (Nanjing UIST)?

Figure 2: What is the base map in the middle of (a) – city lights?

Figure 4: Explain the $\Delta$s – what quantities are subtracted?  Is the PBLH from the model?  Have you compared the simulated PBLH with data?  Are the data plotted in (a) averages for all hours of the day?

Figure 5:  I think the captions for (b) and (c) are switched.

Figure 7:  More explanation is needed in the caption.  What is the origin of the background in (a)?  What are the vertical lines in (b)?  The latter question can be avoided by using the same shading in both panels.

Figure 8:  This figure is good for showing model/observation comparison, but the vertical scales are too compressed to show long-term temporal variations or to compare between years.

Figure 9:  (a) and (b) – the yellow color is very difficult to see.  How do these plots look if you only use mid-day or early-mid afternoon results?

Figure 10: "Observation" in the legend should be plural ("Observations").  What is the solid blue curve in (a) – probably the dashed blue line in the legend.  This is difficult to see.

Figure 11.  The 1:1 lines are not dashed in the figures, although the legends say they are.  Please distinguish the 1:1 lines from the regression lines.

Figure 13.  More explanation is needed in the caption.  "Cement proportion" of what? Total anthropogenic $CO_2$ emissions? EDGAR?  "Cement increase ratios" – please explain what this is. Please be explicit as to what strategies 1 and 2 are, especially since some readers focus on the figures and not on the text.

Table 1.:  Explain "/"

Table 2.:  Can you add rows for the average values for the model results and the observations for both $CO_2$ mixing ratios and $\delta^{13}C$-$CO_2$ for each column?

---

## Referee Comment (RC2) · Anonymous Referee #2 · 16 Mar 2021

Review of Hu et al, ACPD, 2020. "Anthropogenic and natural controls on atmospheric $\delta^{13}$C-$CO_2$ variations in the Yangtze River Delta: Insights from a carbon isotope modeling framework"

General Comments

As the authors point out, this is the first time (that I'm aware of) that $CO_2$ and $\delta^{13}$C have been both modeled and measured for an urban area. I'm glad that they've attempted to tackle this issue, because applying multiple data streams (in this case $CO_2$ and $\delta^{13}$C) within an atmospheric modeling framework should help us better understand urban $CO_2$ sources and sinks. Although the inverse modeling methodology used is somewhat simplistic, I think it's a good start that can later be made more sophisticated. The forward modeling skill is only 'moderate' in my opinion (R^2 < 0.2 for $CO_2$ and marginally better for $\delta^{13}$C), but this is perhaps unsurprising given the high noise urban environment and the relative proximity of the sampling site to local sources. Beyond the simulations, the authors carry out numerous interesting analyses using a combination of model results and observations. The topic they are addressing is important, the study is ambitious in scope, and is completely appropriate for ACP. There is no doubt that this paper represents a great deal of hard work in terms of both measurements, modeling, and analysis. All this said, I have some significant concerns and questions that need to be addressed before the paper can be published. I will outline my concerns immediately below and then provide detailed line by line questions and comments.

1. $\delta^{13}$C data

My biggest concern is with the $\delta^{13}$C measurements presented. In particular, I am having a hard time understanding $\delta^{13}$C values that are greater (more positive) than -6 per mil, at times (Fig. 6), and more generally, summer afternoon values that appear to be close to -7 per mil (Fig. 7h). Looking at well-established background sites in the Northern Hemisphere such as Mauna Loa (from https://scrippsco2.ucsd.edu/data/atmospheric_co2/mlo.html) and La Jolla, $\delta^{13}$C values for July 2015 are around -8.4 and -8.3 per mil, respectively. $\delta^{13}$C from the NOAA network for Dec. 2014 at Mauna Loa (the last month available on their website) is -8.4 vs. -8.6 per mil for Dec. 2014 from the Scripps measurements, strongly suggesting that there are not significant offsets in the Scripps data. Given a rough starting point of ~ -8.4 per mil, it's very difficult to understand how a heavily polluted urban region where, as the authors say, the biosphere is a relatively minor component of fluxes, could raise a broadly representative background value of -8.4 to something close to -7 per mil. In principle, this could occur only with a large removal of $CO_2$ from the atmosphere by net photosynthesis (leaving the atmosphere more enriched). To put a rough number on this, C3 plants fractionate approximately at a ratio of -0.05 per mil/ppm. This means that the $CO_2$ levels seen at this study's measurement site would need to be roughly 28 ppm lower than La Jolla (for example) in July. While it's hard to tell exactly what the July 2015 daytime $CO_2$ is, from Figures 3a and 7g it appears to be right around 400 ppm. For comparison, La Jolla $CO_2$ for the same month is 397 ppm (the CarbonTracker backgrounds mentioned in the paper are very similar to this value). Another possible explanation for high $\delta^{13}$C would be a source of $CO_2$ with an isotopic signature heavier than the atmospheric value of ~ -8 per mil. Cement production ($\delta^{13}$C ~ 0), which is discussed extensively in the text, is of course such a source.

However, even with the relatively large fraction of anthropogenic emissions as cement in the study region, the flux-weighted mean isotopic signature of anthropogenic emissions will still be much lower than the atmospheric values (~ -24 per mil, Fig. 12a). One could also ask the question if air from higher in the atmosphere might have higher $\delta^{13}C$ levels (and be more appropriate to consider as a reference than La Jolla, e.g.). To answer that, we can look at data from 4 km asl from collected aboard aircraft (the NOAA site CAR at around 40 deg. N; see https://www.esrl.noaa.gov/gmd/dv/iadv/), where in July 2015, the $\delta^{13}C$ is around -8.2 per mil, not much different than either La Jolla or Mauna Loa. The NOAA site LEF (a continental forested site with little industrial/urban influence) did record a $\delta^{13}C$ of ~ -7.4 per mil in July, 2015. However, this isotopic enrichment was associated with a $CO_2$ level of ~370 ppm, much lower than the hemispheric mean background. The only time prior to this paper that I've seen such enriched $\delta^{13}C$ values have been in ice core samples (e.g. Francey et al., Tellus, 1999).

So the question is, why is this happening? I am not an expert in optical $\delta^{13}C$ measurements, but reading the referenced paper Xu et al, ACP, 2017 as well as Ghasemifard et al, 2019 (Atmosphere) and Ghasemifard et al, 2019 (Aerosol and Air Quality Research), it is clear that the Picarro instrument used in this study requires significant corrections due to, among other things, water vapor. The fact that the very high values of $\delta^{13}C$ are seen mainly in summer, while in winter (e.g. Fig. S4) the values seem much more reasonable, makes me wonder if the water corrections (which will be much more significant in summer) are playing a role here. It is worth noting here that even with frequent calibrations with reference air of well-assigned $CO_2$ and $\delta^{13}C$, the fact that the reference gas is bone dry while the sample air is moist will pose a problem. I'm not saying here that water vapor is the explanation for the unreasonably enriched values during summer but rather suggesting this as a candidate for investigation. Almost more important than the fact that the data appears to be biased, I'm worried that there may be a seasonally varying bias in the data. Because the seasonality of the signal is an important part of the analysis in the study, it's important to make sure that, at the very least, any biases/offsets in the data are constant. I don't know why the data are as enriched as they are, but it's the authors' responsibility to convincingly explain the high $\delta^{13}C$ values they observe.

2. Daytime analysis

The vast majority of both forward and inverse model analyses have focused on afternoon data. The main reason for this is that atmospheric transport models generally have a much harder time simulating shallow nighttime boundary layers often with strong vertical gradients (where fewer model levels are available to capture vertical gradients) than they do simulating higher mid-day boundary layers, when the PBL tends to be well-mixed. Another big advantage of focusing on mid-day data is that the daytime turbulence in the PBL serves to integrate fluxes over a much larger upwind region (in time and space). Thus, conclusions about sources and sinks, especially when using data from just a single site, are much more likely to be spatially representative. I would like to see the model-data comparisons and other analyses using afternoon selection criteria (12-16 hr, e.g.). Even if model-data comparison statistics do not radically improve, other analyses, such as the enhancement proportions of different sectors could change by minimizing the influence of very local sources. With model-data comparisons, in particular, it could be that the model performs similarly for the full record as it does for just the daytime part. If so, this would be an interesting finding.

3. Equations

As detailed below, the details of the equations are hard to follow, especially in terms of what is simulated and what is measured. I have made some suggestions to clarify the notation.

Specific Comments

L46 change v432 to v4.3.2

L47-48. "and constrained the anthropogenic $CO_2$ emission categories." This is misleading. The scaling factor approach only constrained the total anthropogenic emissions. The isotopic data were used constrain the cement fraction to some extent.

L50. "performed well" This is debatable and subjective. The $R^2$ values for fits to $CO_2$ data were less than 0.2. If you want to comment on WRF performance, please quantify instead of saying "well".

L54. Delta_s has not been defined at this point, so you need to say what it means.

L58. Change 'plants' to 'plant'

L78. Change 'by' to 'from'

L85. I don't think this is the correct IPCC reference. What you want to cite here are the IPCC guidelines on emissions calculations, not the IPCC report on the science of climate change.

L88. Change 'into the inversion of global biological $CO_2$ flux' to 'into the estimation of biological fluxes in atmospheric inversions.' It's not the biological fluxes that are being inverted. Also global is not appropriate here because you are talking about high fossil uncertainties at local scales.

L122. Change 'have recently be' to 'have recently been'

L123. Change 'inversion has been' to 'inversions have been'

L129. Change 'power' to 'the power'

L151. The NOAA/ESRL lab you refer to should now be referred to as NOAA/GML (NOAA Global Monitoring Laboratory).

L176. A) does "ms" in $CO_2\_ms$ refer to measured? Or is the left-hand-side of eq. 1 just a simulated quantity. This seems to be the case from the text above, but to make that clearer, I suggest adopting more intuitive notation such as '$CO_2\_sim$'. B) Delta_CO2 on the right hand side of eq. 1 might be better written as $Sum\{i=i,n\}[Delta\_CO2]i$ to be consistent with eq. 2.

L177.  Change 'hands' to 'hand sides'

L179.  Again, I think the notation should be clarified.  The left hand side of eq. 2, as I understand it, is the simulated value of atmospheric $\delta^{13}C$.  Make that notation consistent with that for the simulated value of atmospheric $CO_2$, e.g. $\delta^{13}C\_sim$.

L183. As mentioned above, in eq. 3, instead of Delta_CO2 use Sum{i=i,n}[Delta_CO2]i.  Then it becomes very clear what the definition of delta_s is: the enhancement-weighted mean isotopic value of all sources/sinks.

L184.  Change 'the mixture' to 'the enhancement-weighted mean', which is more precise.

L194-195. I don't agree with: "background air masses should originate from the free atmosphere at heights of 1000 m or higher above the ground".  In general there shouldn't be a specific altitude requirement for background air.  A more general definition when conducting regional studies would be that background is the concentration or isotope ratio of air when the air enters the regional study domain, which is often determined using back trajectories. The back trajectories (often an ensemble as in the case of STILT) will exit the domain at a variety of altitudes, including possibly below 1000 m.  Also, with regard to WLG in particular, this is a remote high altitude site that would be expected to be sampling free tropospheric air most of the time.  While it is true that WLG is significantly to the west of the domain 1 border, given the size of the $CO_2$ enhancements (and $\delta^{13}C$ depletions) at the observation site, I would expect WLG to be a reasonable background site.  As mentioned above, I think a more likely reason why WLG doesn't appear to be a good background site (more so in summer) has less to do with WLG and more to do with potential bias in the dataset.  One quick experiment you can do is compare the $CO_2$ values you extracted from CarbonTracker with the CarbonTracker values for WLG.  I doubt there will be a substantial difference.  If true, this would suggest that WLG should also be a reasonable background for $\delta^{13}C$.

L198-199. "can cause a high bias in the $\boldsymbol{\delta^{13}}C$-$CO_2$ background when using this approach." Apologies if I am misinterpreting something, but using WLG is much lower (more negative) than the other background approaches.  Why 'high bias'?

L200. 'second approach'.  It seems that the second approach is a very limited approach whose main purpose is to validate the 'third approach'.  Thus, I would move this after the third approach and maybe not call it an 'approach' but say 'in order to test the second approach, we…'.

L203.  Here $\delta^{13}C\_a$ is referred to as an observed quantity, whereas in eq. 2 the same notation was used to refer a simulated quantity.

L205-206. 'minimize simulated $CO_2$ enhancement errors'  Are you referring to errors coming from NEE here? Is that why you chose the wintertime?  If so, state this more explicitly.

L209. Change 'equations' to 'equation'

L209. Is Delta_CO2 here derived from $CO_2\_obs - CO_2\_bg$, or is it simulated? The first usage of Delta_CO2 in eq. 1 implies that Delta_CO2 is a simulated quantity, because you write about eq. 1: "$CO_2$ was simulated as the sum of background ($CO_2\_bg$) and the contribution from all regional sources/sinks ($\Delta CO_2$)". Perhaps eq. 1 is meant to describe that observed $CO_2$ can be decomposed into a background component and the contribution from all sources. (Thus implying that Delta_CO2 is not simulated using footprints, but rather can be derived from observations and the background estimate.) However, on line 210 you write that in the third approach you do not need to simulated Delta_CO2_i. Please clarify.

L213. 'Similar methods…' I wouldn't say the studies referred to use similar approaches. For one, they don't involve isotopes. Second, in most of the approaches referenced, back-trajectories and information from remote sites were combined to determine background. In contrast, and very importantly, the $\delta^{13}C$ background determined using method 3 is not independent of the observations themselves. This is a an important point because this is what allows you to define a background that fits so closely to the upper envelope of the observations, despite the fact that the smooth curve fit through the background (Fig. 6) is close to -6 per mil in the summer of 2015. As mentioned earlier, such values are not physically reasonable.

L221-222. '1000 m above ground' Why wouldn't you use the concentrations from CarbonTracker at the altitudes where the back-trajectories exited the domain, instead of 1000 m agl? This may be a mis-interpretation or mis-reading of the Hu et al. 2019 methodology. As mentioned above, the background concentration should be taken from the lats, lons, and alts at which the ensemble of trajectories exit the domain. These values (500 in the case of STILT?) can then be averaged (and some aspect of their variance, perhaps the std. error of the mean, used to compute an uncertainty.)

L223. 'hourly footprint function' say how these were calculated, or provide a reference here. Also, does this imply that footprints and back-trajectories were calculated for every hour of the data record? If so, state this.

L233. When convolving the fluxes with the footprints to produce Delta_CO2 (here it's usage is clearly as a simulated quantity!), were hourly footprints for a single measurement convolved with hourly NEE to account for covariances in the diurnal patterns of both the footprints and the NEE? In other words, was there just a single footprint per measurement, summed over the 7 days, or were there 7x24 footprints saved? When focusing on simulations of biospheric $CO_2$ this factor is very important. I suspect in your case, where anthropogenic fluxes without a significant diurnal cycle are dominant, neglecting this covariance between NEE and transport is reasonable. However, neither eq. 5 nor the text contain any of this information.

L251. Here, you say that you used CarbonTracker values above 1000 m, which is different than what you said on line 221, where you implied that you said you used values at 1000 m. Please clarify. However, I'll repeat that there is nothing special about 1000 m. You should use the $CO_2$ value from CarbonTracker at whatever altitude the trajectories left the domain.

L254. Say immediately which version of EDGAR you used.

L258.  Modify 'the most up to date global inventory' to 'the most up to date global inventory with sectoral detail'.

L264.  It is not quite true that EDGAR v4.3.2 is only available for 2010.  This is only true for the version with monthly resolution.  Please add this qualifier.

L266.  Please explain more about how the factor of 1.145 was calculated.  For example, was it based just on CT or did it use EDGAR for 2010 and CarbonTracker for 2014 and 2015?  The former would be much better for consistency.

L269.  Change 'posteriori' to 'a posteriori'.

L275.  Again, with notation, please distinguish simulated (or bottom-up) delta_s as in eq. 6 with delta_s determined by a Keeling Plot, e.g.

L280.  How much of non-metallic mineral production is cement?  60%, 90%, 99%?  If nearly all, then I recommend stating this here and saying that from here on you will just refer to it as cement.

L308.  Change 'it' to 'them'

L310.  At least for Hu et al., 2019, the inverse modeling approach was very different than that described here.  It is a bit misleading to cite it.

L313.  Again, confusing notation.  Earlier, $CO_2\_obs$, was referred to as $[CO_2]$.  The "ms" subscript is still a mystery to me.  In eq. 1. $CO_2\_ms$ was total $CO_2$ including all sources (and background).  Here, $Delta\_CO2\_ms$ is only the simulated anthropogenic enhancement.  Why not make the subscript more intuitive, such as '$Delta\_CO2\_anth$'?

L313.  Are these terms monthly means? Or is a scaling factor calculated at high time frequency and then averaged to a monthly mean SF?  Sargent et al., e.g., calculated the mean of all afternoon modeled and observed enhancements.  Also, subsetting to afternoon data only, when modeled PBL heights are likely most reliable, is a necessary test.

L329.  As mentioned in the general comments, evaluating model performance with only afternoon data is strongly advised.

L331.  As mentioned above, I don't agree that $R^2 < 0.2$ equals "good" performance.

L348.  I'm confused as to how you can compare 2014 and 2015, if 2014 was a full year but 2015 was a partial year.  Are you only comparing the months that are common to both years?

L349.  Here you say that the emissions are the same for both years.  This should be mentioned explicitly in the methods section.

L353. I don't think you can conclude that meteorological differences between years were relatively small based on comparison of annual values. More analysis would be needed. For example, there could be a lot of seasonal cancellation that still results in similar annual averages.

L355. In terms of comparing NEE year to year, I think much more analysis is needed for this to be meaningful. First, NEE changes significantly on a seasonal basis. Second, because you are presumably looking at the full record, nighttime respiration is over-represented. Daytime values on the other hand incorporate both daytime NEE (GPP and ER) and respiration from the previous night (and probably more day/night cycles depending on the size of the domain and the windspeeds).

L362. Note that the GPP product used is derived from the SIF product used. They are not really that independent. This can be seen in the shape of their seasonal cycles.

L370. While you may be able to ignore GPP during the winter, it doesn't mean you can ignore respiration. There will still likely be some NEE.

L383. Change Washington to Washington D.C. (to distinguish from Washington state.)

L385. Change 'Eastern' to 'the eastern'

L389. "indicating greater…' Most likely, yes, but also trapping of emissions in the PBL will play a role. You cannot immediately transfer enhancements to emissions. So you need to qualify this by saying something like 'assuming similar windspeeds and PBL heights…'

L392. Say briefly how the percentages were calculated. I assume by convolution of each regions emissions with the footprints.

L401. Change 'oil refinery' to 'oil refineries'

L412. How were the annual scaling factors calculated? As the unweighted (or weighted?) mean of the monthly ones? Also, the MSFs are never presented. What is their seasonality? This seems like a major result of the $CO_2$ part of your analysis (and the 'inversion'). The monthly results should be discussed and/or presented.

L415. Change posteriori to 'a posteriori' and 'for YRD area' to 'for the YRD area'.

L416. The last sentence on cement seems unnecessary and out of place, even if it is true.

L429. Using 'discrimination' to describe the isotope ratio of cement production is not appropriate, because it is not a fractionating process like photosynthesis is, e.g. You could instead say 'the isotopic signature associated with cement production…', e.g.

L431. Change 'plants' to 'plant'

L432. Change 'than observed' to 'than the observed'

L438.  Regarding Chen et al., 2006, the vertical gradients in that study are based on models. Observations were generally only at 20 m agl.  In at least one example (Fig. 1) The vertical gradient looks to just about 0.1 per mil at mid-day.  Moreover, at least for the summer months, the simulated values in the surface layer are more enriched due to photosynthetic drawdown.

L439.  Ghasemifard is not in the reference list.

L440.  Saying that Zugspitze values were around -7 per mil for winter 2013 is not an accurate characterization of the Ghasemifard results.  Looking closely at Fig. 2 of Ghasemifard et al. 2019, the average $\delta^{13}C$ for DJF of 2012/2013 is at least 1 per mil lower than -7.0 per mil (even excluding pollution events).  This extraordinarily high value of ~ -7.0 is only reached in two cases, once in October and once in January.  Also note that -7.0 per mil is an unrealistically high value for Oct. 2012 in relation to similar high altitude data like that from the Scripps Mauna Loa record.

L442.  Saying that a clean air event pulls the $\delta^{13}C$ down from -7 to -7.5 does not make sense. Pollution events ($\delta^{13}C$ of fossil fuels average around -30 per mil) will make $\delta^{13}C$ more negative. Here you are saying a *clean* air event makes the $\delta^{13}C$ more negative.

L449.  As with $CO_2$, doing an afternoon hours-only analysis of model performance would be valuable.

L468.  'and generally caused by…'  you could test this hypothesis by correlating the model minus obs. residuals for CO2 and $\delta^{13}C$.  That is, if $\delta^{13}C$ simulation errors are caused by $CO_2$ errors, there should be a strong correlation and the slope should be related to the mean isotopic signature of the sources.

L471.  In Figure 9, by focusing only on daytime data you may minimize the impact of large night time NEE enhancements (Fig. 7b) and get a better correlation.

L477.  After 'was observed in December and July' reference the relevant figure.

L484.  Regarding delta_s, presumably this was calculated using equation 6 and the emission proportions listed on line 397. And are these the all-day values shown in Figure 12?  Or are these data derived delta_s values from Keeling Plots.  As mentioned earlier, different notation for simulated and data-based delta_s would be helpful.

L496.  The seasonal cycle attribution is very confusing to me. To start with, Fig. 10 shows that about two thirds of the seasonal cycle is from the $\delta^{13}C$ background.  This seems straightforward. However, then you say that the background and regional source terms are 59% and 41% of the seasonal cycle.  And then you say that total $CO_2$ enhancement and $CO_2$ enhancement components further contribute another 20% each.  This seems to add up to 140%.

L499.  I don't think your approach to separately investigate photosynthesis and respiration is valid.  Negative NEE instances will still have a substantial influence from respiration.  And some positive NEE will have substantial photosynthesis.  I highly recommend simplifying the analysis

(and make it more accurate) and only analyze the effect of NEE. You just don't have enough information to estimate the effects separately (unless perhaps you use the GPP estimates to partition NEE).

L506-507 'via respiration' I would change this to 'via net respiration', or 'positive NEE', because photosynthesis might be active at some points.

L525. Eliminate 'a' prior to 0.40 per mil.

L526. Why is the impact of cement expressed as a range?

L542. Another notation question: what exactly is delta_ms? It has not been defined.

L561. It's not clear that earlier in the paper you stated a hypothesis.

L565. I disagree that the cement isotopic signature is the most distinct. In fact it is only ~ 8 per mil away from the atmospheric value, while biological sources and other fossil sources are ~ 20 per mil away and thus should exert more leverage on the atmospheric value. I also disagree that its emission is large. It is large for cement compared to other parts of the world, but it is still only ~ 10%.

L567. Provide a reference for the statement that the YRD is the largest cement producing region in the world.

L588. Regarding the 16.85 %, why is there such a large seasonal cycle in the cement enhancement proportion. Can you discuss this or offer any explanation? Seasonal changes in wind direction? Also, I am a bit confused about the a and b superscripts in Table S2. I thought the cement enhancements were simply calculated by convolving the cement emissions and the footprints. Do a and b just refer to whether the proportion is relative to total flux or only anthropogenic flux? If so, I would not use 'considering' in Table S2 and instead say explicitly that a is ratio of cement/anthro and b is a ratio of cement/total.

L609. I don't think you can conclude that delta_s is more sensitive to cement emissions than other emission categories without testing those categories!

L616. 'contributed 0.40 per mil' to the seasonal cycle? Please say this explicitly.

L631. Change 'write' to 'wrote'

L665. You shouldn't really cite a 'Discussions' paper from 2016. Please replace with the final paper (or remove if not available).

L684. The journal title seems to appear twice.

L845. What is [J]?

L901.  It would help to link panel B to panel A so one can see which of the cement locations are in which domain.  Maybe draw some of the domain boxes in panel B?

L931.  Are the units really nmol/m2/s?  This implies a *maximum* flux of 1e-14 mol/m2/s, based on the colorbar.

L932.  The colorbars/legends are very hard to see in the population density maps.  Also, panel B colorbar needs units.  Also for panel B, which domain in Figure 1 does this concentration footprint correspond to.  It's not clear.

L959.  Consider using something besides 'Delta_CO2' as the y-axis label, because this has already been defined differently elsewhere in the paper.  Also, in the figure caption be more clear about what 'Delta' means, which (I think) are differences between the same monthly averages in two different years.  Panels b,c, and d are referred to after Fig. 5a and are not strongly related to panel a.  I recommend separating them from this figure. Also what does 'distance xx' in the legend mean?

L971.  In panel a should 'model' really be 'anthropogenic'?  b and c are switched in the caption.

L1019.  Do the individual data points in the plots represent daily means?  Please clarify.

L1051.  Relabel panel a y-axis as delta_s.

Note.  There are some language erros/typos in the supplement that need to be fixed.

---

## Author Comment (AC1) · 6 May 2021

*Review of: Anthropogenic and natural controls on atmospheric d13C-CO2 variations in the Yangtze River Delta: Insights from a carbon isotope modeling framework*
*Authors: Cheng Hu, Jiaping Xu, Cheng Liu, Yan Chen, Dong Yang , Wenjing Huang, Lichen, Deng, Shoudong Liu, Timothy Griffis, and Xuhui Lee*

*This paper describes a study of CO2 emissions in the megacity region of the Yangtze River Delta of China, which include several major cities in eastern China. The novel contribution of this study is the WRF-STILT modeling of the emissions making extensive use of the stable isotopic composition of carbon in CO2 (d13C-CO2). The simulation agrees well with the CO2 observations. The modeling of d13C-CO2 allows investigation of the contributions of various anthropogenic and biogenic sources. The topic of this study falls well within the scope of Atmospheric Chemistry and Physics. Therefore, this paper should be published after minor revision.*

We thank the Reviewer for their thoughtful comments and detailed suggestions. All points have been addressed below (review query in Italic; author response in blue). Changes to the text in the manuscript have been marked in bold text.

*My concerns include the need for clarification and further discussion of several points and the need for quantification of uncertainties in calculations resulting from the modeling runs. Particular instances of these are given below.*
Done as suggested.

*Specific comments:*
*Line 114: Replace "be used" with "been used."*
Done as suggested.

*Line 188ff: Move Figure 6 here, since you are describing it here. You should refer to it here, changing the number to 2, and therefore adjusting the figure numbers for the old 2-5 to 3-6, both in the text and in the figure captions.*
Thanks for pointing this out. The description presented here is mainly for the method sections and has not yet referred to the analysis of background data. We believe it is best to keep Figure 6 as is.

*Lines 205-206: The lowest quintile is the lowest 20%, not the lowest 5%. You can just say "the bottom 5%" to describe the data used in this approach to background.*
Done as suggested.

*Lines 216-218: Give the parameters you used in the CCGCRV curve fitting calculations.*
Done as suggested. We added "(a digital filtering curving fitting program developed by the Carbon Cycle Group, NOAA, USA)" following CCGCRV, and also added "we derived CCGCRV curving fitting lines by using 11 regressed parameters, which were based on hourly the time series of observations/simulations".

*Line 231: Add (Figure 2 (perhaps changed to Figure 3)) after "YRD."*
Done as suggested, we added (Figure 2) after YRD.

*Line 243: Replace "East China" with "Eastern China."*
Done as suggested.

*Line 247: Insert "backwards" after "locations."*
Done as suggested.
*Line 250: Replace "at the end of" with "for."*
Done as suggested.

*Lines 263-264 and elsewhere: Replace "EDGAR v432" with "EDGAR v4.3.2."*
Thanks for pointing these out. We have made these corresponding revisions throughout this manuscript.

*Line 276: Please clarify "enhancement." Is this proportion of emissions due to source i?*
*"Enhancement" sounds like it is the amount of CO2 in excess background.*
Here the enhancement is $CO_2$ mixing ratios contributed by different $CO_2$ emission sources as described on lines 180-182 "Note that $\Delta CO_2$ is the sum of all simulated sources/sinks $[\Delta CO_{2\_sim}]_i$ and represents the total simulated $CO_2$ enhancement. We use $\Delta CO_{2\_obs}$ as the observed $CO_2$ total enhancement, which can be calculated by using the $CO_2$ observation minus the $CO_2$ background values.". The enhancement proportion indicates the proportions of a specific enhancement to total $CO_2$ enhancement. We added "where $\delta_i$ is the $\delta^{13}C$-$CO_2$ value from source category $i$, and $p_i$ is the corresponding enhancement proportion (i.e. proportions of a specific enhancement $i$ to total $CO_2$ enhancement). We define $\delta_{s\_sim}$ as the simulated carbon isotope ratio of all sources to differentiate it from the observed $\delta_{s\_obs}$." on lines 301-303.

*Line 286: When considering the biosphere in cities, people are starting to include the effects of*
*human respiration and excretion (Turnbull et al., 2015; Miller et al., 2020, just published in*
*PNAS), using information from Prairie and Duarte (2007). You might want to comment on how*
*this would affect your analysis. The d13C of human respiration should reflect that of the*
*average diet. Also, is any bioethanol used in the gasoline? You should confirm this, since this is*
*common is some cities.*

Thank you for raising this concern. As mentioned in previous studies (Turnbull et al., 2015; Miller et al., 2020), both biofuel combustion and human respiration will emit $CO_2$. For the biofuel combustion related $CO_2$ emissions, there are bioethanol in the gasoline and other fuels, which have been attributed to organic emissions in the EDGAR inventory and considered in our simulations on lines 313-314 as "biofuel combustion and biological emissions ($-28.20‰ \pm 1.00‰$)".

For the $CO_2$ emissions related to human respiration, our previous study found it only accounted for 3.7% of anthropogenic emissions in the YRD area (Xu et al., 2017), which was a relatively smaller role and most of the local human food diet is dominated by $C_3$ grains, having the same $\delta^{13}C$-$CO_2$ value as biological $CO_2$ flux of $-28.20‰$. Also note the biological $CO_2$ flux (used in this study) from Carbon Tracker assimilation system considered anthropogenic is fixed and attributed the rest of $CO_2$ changes to biological $CO_2$ flux (Peters et al., 2007). Therefore we believe the uncertainty of the biological $CO_2$ flux will contain the small proportion of human respiration. We have added more description to clarify it on lines 316-323 "Since $CO_2$ emissions associated with human respiration (Prairie and Duarte, 2017; Turnbull et al., 2015; Miller et al., 2020), is relatively small (3.7% of anthropogenic emissions in the

YRD area, Xu et al., 2017), and given that the local food diet is dominated by $C_3$ grains that have a similar $\boldsymbol{\delta}^{13}$C-CO$_2$ value as the biological CO$_2$ flux of −28.20‰, we assume it has the same isotope signals as local $C_3$ plants and ecosystem respiration. Further, the biological CO$_2$ flux from the Carbon Tracker assimilation system considered anthropogenic as fixed and attributed the remainder to the biological CO$_2$ flux (Peters et al., 2007). Consequently, we believe the uncertainty in the biological CO$_2$ flux will include the small proportion of human respiration."

*Line 322: Replace "blue" with "blue-red."*
Done as suggested.

*Lines 333-334: Replace "below" with "to."*
Done as suggested.

*Line 335: What hours did you use for daytime? Most modelers stress that mid-day to midafternoon hours work best, when the planetary boundary layer height (PBLH) is best simulated.*
Here we displayed (1) all hours (2) nighttime and (3) daytime, respectively in Figure 3, and hours between 10:00 to 16:00 represent daytime, hours between 22:00-6:00 are for nighttime, we added them on line 371 for clarification, which was also list in the label of Figure 3. We agree that generally daytime hours work better for the PBLH variations because of strong vertical development in the daytime, meanwhile some recent studies also found WRF-STILT will underestimated the PBLH compared with Lidar observations (Sargent et al., 2018), which indicate the daytime PBLH performance is not as good as nighttime. Another reason for choosing all-day simulations is that both biological and anthropogenic CO$_2$ flux have strong diurnal variations (i.e. much higher in daytime and lower in nighttime for anthropogenic emissions), so if only use daytime observations, the derived scaling factors will reflect bias in both the *a priori* diurnal scaling factors and daily averages of CO$_2$ emissions, so even the scaling factors is larger than 1, it does not only indicate the anthropogenic CO$_2$ is underestimated, it can also be caused by underestimation of diurnal scaling factors in daytime not the daily averages. Considering above factors, we compared model-observation for daytime, nighttime and whole day in Figure 3b-d, and finally choose the whole day averages to scale monthly CO$_2$ emissions.

[Figure]

Figure R1. Derived monthly scaling factors for all day and only day time.

We added the following details: "The monthly scaling factors derived from using daytime and all-day observations are also shown in Figure S4. These factors vary seasonally with higher values observed in summer. When using daytime values only, the scaling factors were much larger than the all-day values. This can be seen in Figure 3 by comparing the simulated and observed $CO_2$ mixing ratios. We should note here that the larger scaling factors based on the daytime data could be caused by bias in the *a priori* daily scaling factors used to generate the hourly $CO_2$ emissions (Hu et al., 2018b); the monthly anthropogenic averages; and bias in negative biological $CO_2$ enhancement. Since our study is mainly focused on the seasonality of all-day observations, the monthly scaling factors derived from the all-day approach will be used for the following analyses." on lines 455-464 for clarification.

*Your Figure 3 suggests that the monthly average of nighttime modeling results matches the observation best.*
Yes. The results indicate that nighttime modeling has less bias than daytime. It's one of the reasons why we not choose all-day instead of daytime observations to do the constraint at monthly scales.

*Line 345: What are the two months that fall far below the trend in Figure 4a? Do you have an explanation for these?*
These two months are March and August. It indicates that PBLH variations and other meteorological factors (i.e. monthly changed footprints sources) also play a role in affecting $CO_2$ variations. We added on lines 383-385 "We also note that there were two months (March and August) that fall far below this trend, implying that changes in the monthly footprints (source area) can also play an important role."

*Line 359: Neither Figure 5a nor 7b is consistent with a negative average summer NEE. Indeed, Figure 5a suggests the opposite since all 2014 summer months are positive in NEE/biological contribution to the CO2 enhancement, as are June and July 2015.*
Yes, as displayed in Figure 7b, the daytime NEE are generally negative and nighttime NEE are positive in summers, which will lead to negative $CO_2$ enhancement in daytime and positive $CO_2$ enhancement in nighttime, respectively. The Figure 5a displayed monthly averaged biological $CO_2$ enhancement which will smooth the diurnal variations. Another reason is that daytime PBLH were generally much higher than nighttime, which leads to much lower absolute $CO_2$ enhancement in daytime than in nighttime, and the averages of daily or monthly $CO_2$ enhancement will appeared as $CO_2$ positive values.

*Line 363: Replace "4b-c" with "4b-d."*
Done as suggested.

*Lines 375-376: What are the uncertainties in the observed anthropogenic CO2 enhancements? In general,* please give uncertainties.
Here, uncertainty of the observed anthropogenic $CO_2$ enhancement mainly came from $CO_2$ background and simulated biological enhancement. Where both $CO_2$ background and biological NEE were derived from Carbon Tracker systems as described in Methods Section. To quantify the uncertainty of Carbon Tracker $CO_2$ background data, we first calculated the annual averages at Mauna Loa background site (https://scrippsco2.ucsd.edu/data/atmospheric_co2/mlo.html, the red dot as list below). The averages were 398.04 ppm and 400.08 ppm, and the background values derived from Carbon Tracker system were 400.43 ppm and 402.21 ppm for 2013 and 2014, respectively. The comparisons between the Mauna Loa

site and Carbon Tracker system were consistent with only a 2 ppm difference. We believe the actual bias in the Carbon Tracker system above China is likely smaller than 2 ppm because the atmosphere should be slightly enhanced by local emissions compared to Mauna Loa site. Based on the above analyses, we attribute a 2 ppm uncertainty to the background estimate. For the uncertainty derived from simulated biological enhancement, we attribute a larger 50% relative bias based on our previous study (Hu et al., 2018b), which used eddy covariance flux measurement to evaluate biological $CO_2$ flux in Carbon Tracker systems. Based on the above calculations, we updated the results on lines 413-416 as:

"were 38.36±3.32 ppm and 37.89±2.80 ppm for 2014 and 2015, respectively. Here, the uncertainty of the observed anthropogenic $CO_2$ enhancements was calculated by prescribing a 2 ppm potential bias for the Carbon Tracker $CO_2$ fields and 50% to the simulated biological $CO_2$ enhancement (Hu et al., 2018b)."

[Figure]

Figure R2. Locations of Mauna Loa background site (red color).

*Lines 388-390: The absolute enhancements depend on many things, including the meteorology and the magnitude of the emissions. You can't conclude that the YRD has more emissions simply because the enhancements are higher. Modeling is critical for coming to that conclusion.*

Thank you for pointing this out. We agree that $CO_2$ enhancements are influenced by meteorology and the magnitude of the emissions. We deleted "indicating greater anthropogenic $CO_2$ emission".

*Line 392: Explicitly explain where these percentages come from.*

These percentages were calculated by dividing the simulated $CO_2$ enhancement from each province by simulated total $CO_2$ enhancement for the whole domain. The $CO_2$ enhancement from each province was simulated by multiplying $CO_2$ emissions in each province with the corresponding footprint. For clarification, we added "The $CO_2$ enhancements from each of the 5 zones were simulated by multiplying $CO_2$ emissions in each province with the corresponding footprint." on lines 257-259.

*Line 395: Where do you show that the maximum source contribution exceeded 50% on 19 September 2013?*

The reason to mention this extreme situation to illustrate the large influence of long-distance transport at some special periods. We added "not shown" for clarification on line 435.

*Lines 396-399: Please explain how the "anthropogenic enhancement" is different from the*

*"anthropogenic emissions."*

The anthropogenic emissions represent anthropogenic $CO_2$ emissions (or flux), and anthropogenic enhancement represents anthropogenic $CO_2$ enhancement (or concentration) simulated by using $CO_2$ emissions in atmospheric transport model. Theoretically, if $CO_2$ emissions for different categories were homogeneously distributed, the two proportions of "anthropogenic enhancement" and "anthropogenic emissions" for the same category should be the same, while in the real situations both meteorological factors and $CO_2$ emission's spatial distributions will bring inconsistence between the "anthropogenic enhancement" and "anthropogenic emissions" for each $CO_2$ category. The comparisons between them is to illustrate whether enhancement proportions between each $CO_2$ category can represent corresponding emission proportions. We added "The comparisons between the proportions of simulated enhancement and proportions of corresponding $CO_2$ emissions can illustrate whether $CO_2$ enhancement partitions is a good tracer for emissions in a complex urban area." on lines 439-441.

*Line 408: Replace "2014" with "2014-2015."*
Done as suggested.

*Lines 408 and 415: Express the emissions as *1011 kg, the same units as in Table 1, for consistency.*
Done as suggested. We also changed the units from $10^{12}$ kg to $10^{11}$ kg on line 451.

*Line 417: Replace "is" with "are."*
Done as suggested.

*Lines 430-431: A positive biological CO2 signal during winter is consistent with a negligible role for photosynthesis, but it could be that photosynthesis is still important, just not as important as respiration. Will human respiration affect this?*
Yes. We agree that it can be also explained by the fact that photosynthesis is still important, just not as important as respiration. These changes have been applied to lines 487-488 as "which implies a positive biological $CO_2$ signal where ecosystem respiration is more important than photosynthesis"
As replied above for the $CO_2$ emissions related to human respiration, our previous study found it only accounted for 3.7% of anthropogenic emissions in the YRD area (Xu et al., 2017), which was a relatively smaller role and most of the local human food diet is dominated by $C_3$ grains, having the same $\boldsymbol{\delta}^{13}$C-$CO_2$ value as biological $CO_2$ flux of −28.20‰. Also note the biological $CO_2$ flux (used in this study) from Carbon Tracker assimilation system considered anthropogenic is fixed and attributed the rest of $CO_2$ changes to biological $CO_2$ flux (Peters et al., 2007). Therefore we believe the uncertainty of the biological $CO_2$ flux will contain the small proportion of human respiration. We have added more description to clarify it on lines 316-323 "Since $CO_2$ emissions associated with human respiration (Prairie and Duarte, 2017; Turnbull et al., 2015; Miller et al., 2020), is relatively small (3.7% of anthropogenic emissions in the YRD area, Xu et al., 2017), and given that the local food diet is dominated by $C_3$ grains that have a similar $\boldsymbol{\delta}^{13}$C-$CO_2$ value as the biological $CO_2$ flux of −28.20‰, we assume it has the same isotope signals as local $C_3$ plants and ecosystem respiration. Further, the biological $CO_2$ flux from the Carbon Tracker assimilation system considered anthropogenic as fixed and attributed the remainder to the biological $CO_2$ flux (Peters et al., 2007). Consequently, we believe the uncertainty in the biological $CO_2$ flux will include the small proportion of human respiration."

*Line 434: Replace "domain" with "background."*
Done as suggested.

*Line 435: Add "(Figure 6)" after "respectively." This may become Figure 2.*
Done as suggested.

*Line 454: Replace "Figures 3 and 7" with "Figures 3a and 8", but the vertical scale in Figure 8 is too compressed to be seen clearly.*
Done as suggested.

*Lines 465-469: Please give uncertainties. Are the seasonal increases significant? Could PBLH simulation issue explain the large discrepancies, especially since the model diurnal variations are greater than those in the observations.*
The main uncertainties associated with the simulation of hourly $CO_2$ are uncertainty in the meteorological fields, transport model, and *a priori* $CO_2$ flux. As shown in Figure R3, linear relationship between hourly $CO_2$ and $\delta^{13}C$-$CO_2$ bias were observed. This suggests the hourly $\delta^{13}C$-$CO_2$ simulations have similar bias as the sources. At the annual scale, the main uncertainty for both $CO_2$ and $\delta^{13}C$-$CO_2$ is attributed to the PBLH simulations and *a priori* anthropogenic $CO_2$ emissions. Here the bias for *a priori* anthropogenic $CO_2$ emissions were < 6% as calculated in this study, and the bias caused by PBLH uncertainty was usually <13% (Hu et al., 2018a; 2018b). Therefore, we attribute an uncertainty of 20% for the simulated $CO_2$ and $\delta^{13}C$-$CO_2$ at the annual scale. We have added "The main uncertainties associated with the simulation of hourly $CO_2$ and $\delta^{13}C$-$CO_2$ are uncertainty in meteorological fields, transport model (i.e. number of released particles), and *a priori* $CO_2$ fluxes. At the annual scale the main uncertainty is attributed to the PBLH simulations and *a priori* anthropogenic $CO_2$ emissions. The anthropogenic $CO_2$ emissions biases were < 6% as described above, and the bias associated with PBLH uncertainty was typically <13% (Hu et al., 2018a; 2018b). There, we attribute a 20% uncertainty to the simulated $CO_2$ and $\delta^{13}C$-$CO_2$ signals on an annual time scale." See lines 469-475 for clarification.

[Figure]

Figure R3. Relationship of observation minus simulation residual between $CO_2$ and $^{13}CO_2$ for (a) winter in 2013, (b) summer in 2014, (c) winter in 2014, and (d) summer in 2015.

We changed "significantly" with "obviously" for clarification. We also agree that the large discrepancies of hourly $\delta^{13}$C-CO$_2$ variations were mainly caused by CO$_2$ simulations, which was basically caused by simulations of atmospheric transport process and PBLH simulations can have large influence. We added "(potentially caused by PBLH simulation issue during these periods)" on line 526-527.

*Line 505: Replace "than" with "in." The baseline simulation in Figure 10b (red) is more*
*enriched in the heavy isotope, as evidenced by its less negative values between April and*
*October.*
Done as suggested. We revised the typo by replacing "than" with "in". Yes. The baseline simulations of $\delta^{13}$C-CO$_2$ in Figure 10b (red line, containing photosynthesis) is more enriched in heavy $\delta^{13}$C-CO$_2$ than blue line (excluding photosynthesis) between April and October, which was caused by discrimination associated with ecosystem photosynthesis as previously explained on lines 485-487.

*Lines 509ff: Are the differences of 0.08-0.20‰ significant? Please give uncertainties. Similar*
*comment for the next paragraph.*
The difference of 0.08-0.20‰ only accounted for 5%~13% of observed/simulated $^{13}$CO$_2$ seasonality ~1.5‰. Since there are only 2 numbers, the statistics cannot be calculated to report significance. We revised this sentence as "Generally, both ecosystem photosynthesis and respiration played minor roles in controlling the atmospheric $\delta^{13}$C-CO$_2$ seasonality within this urban area".

*Line 530: Add the definition of the regional source term "(ds*DCO2)."*
*In general, be consistent with "ds"*
The $\delta$s$\times\Delta$CO$_2$ can be treated as the regional source term. For additional clarification we have added "The product on the right-hand side of equation 3 is the simulated regional source term that is added to the background value and contains both enhancement and $\delta^{13}$C-CO$_2$ signals contributed by different CO$_2$ sources/sinks. This product can also be treated as an observed term when using the derived $\delta_{s\_obs}$ and observed $\Delta$CO$_{2\_obs}$ values" on lines 192-195.

*Line 544: If you use nighttime simulations, you still have respiration.*
Yes. We agree that nighttime observations will still include respiration. The choice of choosing nighttime data is to minimize the influence of respiration and to mainly focus on anthropogenic CO$_2$ sources. We have added "mainly" before "focus on the anthropogenic CO$_2$ sources" on line 614.

*Lines 597-598: Do you need to show both 13a and b? They are almost identical. You could just*
*show one and generally state the results for the second in the text.*
Since both nighttime and all-day were analyzed (see response to Reviewer 2) we prefer to retain these analyses.

*Lines 600-601: Replace "relatively similar with" with "similar to."*
Done as suggested.

*Lines 608-609: Insert "absolute" after "1.57‰." Are the uncertainties in your calculations small enough*
*that "a 0.013‰ – 0.038‰ change" is significant?*

Done as suggested. The uncertainty of the calculated sensitivity of change in atmospheric $\delta_{13}$C-CO$_2$ to cement proportions should be much smaller than "0.013‰ – 0.038‰" because the uncertainty is a relative value not an absolute value.

*Lines 621-622: Add "calculated from the simulations" before "was shown" and "From the EDGAR v4.3.2 inventory" after "proportion."*
Done as suggested.

*Figure 1: More information is needed in the caption – significance of the different-colored boundaries. Is the red triangle in (b) the same as the blue dot in (a) (Nanjing UIST)?*
Done as suggested. We revised the caption as "Figure 1. (a) Weather Research and Forecasting Model simulation domains and the location of WLG site , the different region colors represent three domains, (b) cement production distribution in YRD and Eastern China. Both green dot in (a) and red star in (b) are NUIST observation site."

*Figure 2: What is the base map in the middle of (a) – city lights?*
The base map is annual total anthropogenic CO$_2$ emissions in our study domains, and it is explained in the caption.

*Figure 4: Explain the Ds – what quantities are subtracted? Is the PBLH from the model? Have you compared the simulated PBLH with data? Are the data plotted in (a) averages for all hours of the day?*
The $\Delta$PBL height is the difference of simulated PBL heights in the same month for different years. Since there is a lack of PBLH observations, it has not been compared to field observations. The data in (a) are for all hours. We also added "these data points represent the difference of monthly averages in two different years for all hours." in the caption of Figure 4.

*Figure 5: I think the captions for (b) and (c) are switched.*
Done as suggested.

*Figure 7: More explanation is needed in the caption. What is the origin of the background in (a)? What are the vertical lines in (b)? The latter question can be avoided by using the same shading in both panels.*
Done as suggested.

*Figure 8: This figure is good for showing model/observation comparison, but the vertical scales are too compressed to show long-term temporal variations or to compare between years.*
Done as suggested, we have expanded the y-axis of CO$_2$ and $\delta^{13}$C-CO$_2$.

*Figure 9: (a) and (b) – the yellow color is very difficult to see. How do these plots look if you only use mid-day or early-mid afternoon results?*
Done as suggested, we changed yellow to red color.

[Figure]

Figure R4. Scatter plots of observed versus modeled (a) winter time $CO_2$ mixing ratios, (b) winter time $\delta^{13}$C-$CO_2$, (c) summer time $CO_2$, and (d) summer time $\delta^{13}$C-$CO_2$ for both years, here these dots are day-time (10:00-16:00) averages.

We also did the comparisons by only choosing daytime observations. The results indicated that daytime $CO_2$ mixing ratio simulations in summer were slightly underestimated and that this causes $\delta^{13}$C-$CO_2$ to be overestimated. The simulations in winter can generally capture the trends for both $CO_2$ and $\delta^{13}$C-$CO_2$, during which the biological $CO_2$ enhancement played a relatively smaller role than anthropogenic emissions. We added this figure in supplemental file and discussed it in main text on lines 533-542.

*Figure 10: "Observation" in the legend should be plural ("Observations"). What is the solid blue curve in (a) – probably the dashed blue line in the legend. This is difficult to see.*
Done as suggested.

*Figure 11. The 1:1 lines are not dashed in the figures, although the legends say they are. Please distinguish the 1:1 lines from the regression lines.*
Done as suggested.

*Figure 13. More explanation is needed in the caption. "Cement proportion" of what? Total anthropogenic CO2 emissions? EDGAR? "Cement increase ratios" – please explain what this is. Please be explicit as to what strategies 1 and 2 are, especially since some readers focus on the figures and not on the text.*
Done as suggested. We revised this caption as "Sensitivity tests showing the influence of cement $CO_2$ emissions on $\delta_s$ for (a) nighttime, (b) all-day, and (c) the relation between cement $CO_2$ and $\delta^{13}$C for simulation strategies 1 (There is no bias in the total anthropogenic $CO_2$ enhancement such that a proportional increase/decrease in the cement component does not change the relative anthropogenic contributions) and 2 (only the cement enhancement changes). Note that the numbers in brackets indicate changes in $\delta^{13}$C with cement $CO_2$ enhancement proportion (the fraction of cement $CO_2$ enhancement to

simulated total $CO_2$ enhancement) increase by 0.2 times. The x-axis values indicate changing cement enhancement proportions to 0.8 1.2, 1.4, 1.6, 1.8, and 2 times the original values." for clarification.

*Table 1.: Explain "/"*
Done as suggested, the "/" means not available.

*Table 2.: Can you add rows for the average values for the model results and the observations for both CO2 mixing ratios and d13C-CO2 for each column?*

Done as suggested.

---

## Author Comment (AC2) · 6 May 2021

*Review of Hu et al, ACPD, 2020. "Anthropogenic and natural controls on atmospheric δ13C-CO2 variations in the Yangtze River Delta: Insights from a carbon isotope modeling framework"*

*General Comments*

*As the authors point out, this is the first time (that I'm aware of) that CO2 and δ13C have been both modeled and measured for an urban area. I'm glad that they've attempted to tackle this issue, because applying multiple data streams (in this case CO2 and δ13C) within an atmospheric modeling framework should help us better understand urban CO2 sources and sinks. Although the inverse modeling methodology used is somewhat simplistic, I think it's a good start that can later be made more sophisticated. The forward modeling skill is only 'moderate' in my opinion (R^2 < 0.2 for CO2 and marginally better for δ13C), but this is perhaps unsurprising given the high noise urban environment and the relative proximity of the sampling site to local sources. Beyond the simulations, the authors carry out numerous interesting analyses using a combination of model results and observations. The topic they are addressing is important, the study is ambitious in scope, and is completely appropriate for ACP. There is no doubt that this paper represents a great deal of hard work in terms of both measurements, modeling, and analysis. All this said, I have some significant concerns and questions that need to be addressed before the paper can be published. I will outline my concerns immediately below and then provide detailed line by line questions and comments.*

We thank the reviewer for the positive comments and detailed suggestions. We have made extensive revisions based on these comments.

*1. δ13C data*

*My biggest concern is with the δ13C measurements presented. In particular, I am having a hard time understanding δ13C values that are greater (more positive) than -6 per mil, at times (Fig. 6), and more generally, summer afternoon values that appear to be close to -7 per mil (Fig. 7h). Looking at well-established background sites in the Northern Hemisphere such as Mauna Loa (from https://scrippsco2.ucsd.edu/data/atmospheric_co2/mlo.html) and La Jolla, δ13C values for July 2015 are around -8.4 and -8.3 per mil, respectively. δ13C from the NOAA network for Dec. 2014 at Mauna Loa (the last month available on their website) is -8.4 vs. -8.6 per mil for Dec. 2014 from the Scripps measurements, strongly suggesting that there are not significant offsets in the Scripps data. Given a rough starting point of ~ -8.4 per mil, it's very difficult to understand how a heavily polluted urban region where, as the authors say, the biosphere is a relatively minor component of fluxes, could raise a broadly representative background value of -8.4 to something close to -7 per mil. In principle, this could occur only with a large removal of CO2 from the atmosphere by net photosynthesis (leaving the atmosphere more enriched). To put a rough number on this, C3 plants fractionate approximately at a ratio of -0.05 per mil/ppm. This means that the CO2 levels seen at this study's measurement site would need to be roughly 28 ppm lower than La Jolla (for example) in July. While it's hard to tell exactly what the July 2015 daytime CO2 is, from Figures 3a and 7g it appears to be right around 400 ppm. For comparison, La Jolla CO2 for the same month is 397 ppm (the CarbonTracker backgrounds mentioned in the paper are very similar to this*

*value). Another possible explanation for high δ13C would be a source of CO2 with an isotopic signature heavier than the atmospheric value of ~ -8 per mil. Cement production (δ13C ~ 0), which is discussed extensively in the text, is of course such a source. However, even with the relatively large fraction of anthropogenic emissions as cement in the study region, the flux-weighted mean isotopic signature of anthropogenic emissions will still be much lower than the atmospheric values (~ -24 per mil, Fig. 12a). One could also ask the question if air from higher in the atmosphere might have higher δ13C levels (and be more appropriate to consider as a reference than La Jolla, e.g.). To answer that, we can look at data from 4 km asl from collected aboard aircraft (the NOAA site CAR at around 40 deg. N; see https://www.esrl.noaa.gov/gmd/dv/iadv/), where in July 2015, the δ13C is around -8.2 per mil, not much different than either La Jolla or Mauna Loa. The NOAA site LEF (a continental forested site with little industrial/urban influence) did record a δ13C of ~ -7.4 per mil in July, 2015. However, this isotopic enrichment was associated with a CO2 level of ~370 ppm, much lower than the hemispheric mean background. The only time prior to this paper that I've seen such enriched δ13C values have been in ice core samples (e.g. Francey et al., Tellus, 1999).*

*So the question is, why is this happening? I am not an expert in optical δ13C measurements, but reading the referenced paper Xu et al, ACP, 2017 as well as Ghasemifard et al, 2019 (Atmosphere) and Ghasemifard et al, 2019 (Aerosol and Air Quality Research), it is clear that the Picarro instrument used in this study requires significant corrections due to, among other things, water vapor. The fact that the very high values of δ13C are seen mainly in summer, while in winter (e.g. Fig. S4) the values seem much more reasonable, makes me wonder if the water corrections (which will be much more significant in summer) are playing a role here. It is worth noting here that even with frequent calibrations with reference air of well-assigned CO2 and δ13C, the fact that the reference gas is bone dry while the sample air is moist will pose a problem. I'm not saying here that water vapor is the explanation for the unreasonably enriched values during summer but rather suggesting this as a candidate for investigation. Almost more important than the fact that the data appears to be biased, I'm worried that there may be a seasonally varying bias in the data. Because the seasonality of the signal is an important part of the analysis in the study, it's important to make sure that, at the very least, any biases/offsets in the data are constant. I don't know why the data are as enriched as they are, but it's the authors' responsibility to convincingly explain the high δ13C values they observe.*

We thank the reviewer for the very thorough comments and insights regarding the $\delta^{13}$C signal. Here we discuss the main reasons for the relatively high $\delta^{13}$C values and provide a rational for our revised approach.

1. **$\delta^{13}$C-CO$_2$ background observations**
Recently, Ghasemifard et al. (2019) showed that hourly $\delta^{13}$C-CO$_2$ values at Mount Zugspitze, the highest (2650 m) mountain in Germany, varied between -7‰ and -12‰ in the winter for 2013. During two especially clean air events (in October and February) at Mount Zugspitze, the $\delta^{13}$C-CO$_2$ was approximately -7‰, during which the CO$_2$ mixing ratios varied between 390 and 395 ppm. This is consistent with our estimates using strategies 2 and 3. We have

added it on lines 497-501.

2. Noise in the empirical estimate of the background values

In Figure 6, the hourly $\delta^{13}$C-$CO_2$ background values were derived by combining hourly $\delta^{13}$C-$CO_2$ observations with the Miller-Tans approach. This derived result is subject to large fluctuations at the hourly time scale because in equation 4, the derived $\delta^{13}$C-$CO_2$ background will have similar hourly variations with the atmospheric $\delta^{13}$C-$CO_2$ observations. For example, the derived hourly $\delta^{13}$C-$CO_2$ background values fluctuated by more than 2‰ within a single day (shown in Figure 6). These large fluctuations are physically unrealistic given our understanding of background values observed at remote sites. For these reasons, we used the smoothing and fitting technique to provide a best estimate of the slow varying background component.

3. The effects of water vapor on the $\boldsymbol{\delta}^{13}$C-$CO_2$ IRIS measurements

As described in our previous work (Xu et al., 2017), we found that the Picarro IRIS $\boldsymbol{\delta}^{13}$C-$CO_2$ measurement has some dependence on the ambient water vapor mixing ratios. This bias was quantified based on sensitivity analyses of increasing water vapor mixing ratios while measuring an air cylinder with a constant $\boldsymbol{\delta}^{13}$C-$CO_2$ value. We observed an increasing trend of 0.46‰ of measured $\boldsymbol{\delta}^{13}$C-$CO_2$ for a 1% increase in water vapor when its mixing ratios exceeded a value of 2.03%. This dependence is shown below in Figure R1.

[Figure]

Figure R1. Dependence of the observed $\delta^{13}$C on the $H_2O$ mole fraction. The lines represent error bars are ±1 SD of 1 min averages. The data in the left panel were obtained on 1 October 2014 using a 439 μmol mol$^{-1}$ standard gas cylinder with a true $\delta^{13}$C value of −32.8 ‰. The right-hand panel is for 10 June 2015 using a 488 μmol mol$^{-1}$ standard gas and the true $\delta^{13}$C value of −34.1 ‰.

The correction procedure is as follows:
$$\delta^{13}C = \delta^{13}C_{true}, \quad C(H_2O) \leq 2.03\%,$$
$$\delta^{13}C = \delta^{13}C_{true} + 0.46‰, \quad (C(H_2O)\% - 2.03\%),$$
$$C(H_2O) > 2.03\%,$$
where the $\delta^{13}$C$_{true}$ is the true isotope delta value, $\delta^{13}$C is the measured isotope delta value (after a two-point calibration), and C($H_2O$) is water vapor mole fraction. This sensitivity test indicates that the true $\delta^{13}$C value is not sensitive to water vapor when water vapor mole fraction is lower than 2.03% (Figure R1, R2). The $\delta^{13}$C value is biased high when water vapor mole fraction is higher than 2.03%, which should be corrected following this calibration

procedure. Based on above equations and our observed values during this study we find that the $\delta^{13}$C values will not be subject to this calibration in the winter and the highest hourly corrections were 0.74‰ as observed during the summer when water vapor mole fraction is higher (Figure R2b).

[Figure]

Figure R2. (a) Hourly variations of observed $H_2O$ mixing ratio, and (b) corrections on hourly $\boldsymbol{\delta}^{13}$C-CO$_2$ observations.

We added the following description "We note that the $\boldsymbol{\delta}^{13}$C-CO$_2$ IRIS (model G1101-i) measurements are sensitive to water vapor concentration. Sensitivity tests reveal that the $\boldsymbol{\delta}^{13}$C-CO$_2$ IRIS measurements are biased high when water vapor mole fraction exceeds 2%. The data presented here have been corrected following the procedures outlined in Xu et al. (2017)." on lines 153-156 for clarification.

4.  Other biophysical factors
We agree that three other potential reasons can help to explain the enriched (or high) $\boldsymbol{\delta}^{13}$C-CO$_2$ values as discussed on lines 201-210 and 485-488 including: (1) vertical gradients of $\boldsymbol{\delta}^{13}$C-CO$_2$; (2) fractionation associated with ecosystem photosynthesis; and (3) enrichment associated with the CO$_2$ derived from cement production.

5.  Our best estimate of the background $\delta^{13}$C-CO$_2$ values

Although the derived hourly $\delta^{13}$C-CO$_2$ background values fluctuated by more than 2‰ within a single day, here we also calculated the daily minimum background $\delta^{13}$C-CO$_2$ in winter, which is displayed with yellow line in Figure R3, the average was -8.15‰ and comparable with WLG winter observations of -8.55‰. Based on the above analyses and discussion, we believe that our best estimate of the $\delta^{13}$C-CO$_2$ background values in the WRF-STILT model framework are derived from smooth curve fitting.

[Figure]

Figure R3. Comparisons among three strategies for calculating the background $\delta^{13}$C-CO$_2$, with figure caption the sane as Figure 6 in the main text.

2. *Daytime analysis*

*The vast majority of both forward and inverse model analyses have focused on afternoon data. The main reason for this is that atmospheric transport models generally have a much harder time simulating shallow nighttime boundary layers often with strong vertical gradients (where fewer model levels are available to capture vertical gradients) than they do simulating higher mid-day boundary layers, when the PBL tends to be well-mixed. Another big advantage of focusing on mid-day data is that the daytime turbulence in the PBL serves to integrate fluxes over a much larger upwind region (in time and space). Thus, conclusions about sources and sinks, especially when using data from just a single site, are much more likely to be spatially representative. I would like to see the model-data comparisons and other analyses using afternoon selection criteria (12-16 hr, e.g.). Even if model-data comparison statistics do not radically improve, other analyses, such as the enhancement proportions of different sectors could change by minimizing the influence of very local sources. With model-data comparisons, in particular, it could be that the model performs similarly for the full record as it does for just the daytime part. If so, this would be an interesting finding.*

Done as suggested. Here we chose to display the results of daytime (10:00-16:00, local time) to represent the periods with higher mid-day boundary layers. We compared the time series of CO$_2$ mixing ratios, enhancement proportions of different sectors, scatter plots, and monthly scaling factors as described below.

[Figure]

Figure R4. Comparisons of $CO_2$ daily averages for (a) all-day and (b) daytime.

We calculated the RMSE, R and Mean Bias (MB) for daily averages for all-day and daytime only, which were 18.68 ppm, 0.44 and 2.68 ppm for all day averages, and 25.21 ppm, 0.38 and 10.74 ppm for day time.

[Figure]

Figure R5. Components of anthropogenic sources for (a) daytime (10:00-16:00), (b) nighttime (22:00-06:00), and (c) all-day (0:00-24:00).

We also re-examine the comparisons of atmospheric $CO_2$ components for daytime (10:00-16:00) and the nighttime (22:00-06:00) results. Figure R5 indicated that they have similar trends, but with different magnitude. We attribute this to differences in the source footprint for day vs nighttime.

[Figure]

Figure R6. Scatter plots of observed versus modeled (a) winter time $CO_2$ mixing ratios, (b) winter time $\delta^{13}C$-$CO_2$, (c) summer time $CO_2$, and (d) summer time $\delta^{13}C$-$CO_2$ for both years, here these dots are day-time (10:00-16:00) averages.

We also performed comparisons by only choosing the daytime observations. The results indicated that daytime $CO_2$ mixing ratio simulations in the summer were slightly underestimated. This caused $\delta^{13}C$-$CO_2$ to be overestimated (Figure R6). The simulations for winter generally captured the trends for both $CO_2$ and $\delta^{13}C$-$CO_2$ when the biological $CO_2$ enhancement played a relatively small role compared to anthropogenic emissions. The larger bias in the summer could result from the relatively coarse spatial-temporal resolution (aggregation error) of the Carbon Tracker biological $CO_2$ flux, which was $1\times1$ degree with three-hour average. As shown in Figure S3, the spatial distribution of land use is far more heterogeneous. This will smooth the stronger biological $CO_2$ signals by averaging it over the large $1\times1$ degree grid, while the urban biological $CO_2$ flux occurs at much finer spatial scales and likely varies at shorter time intervals. We add this description on lines 529-542.

[Figure]

Figure R7. Derived monthly scaling factors for all day and only day time.

The monthly scaling factors derived by only using daytime $CO_2$ observations are displayed in Figure R7. We should note that they are pretty close to results with using all-day $CO_2$ observations from October to March, when the biological flux is smaller compared to the main growing season. During April to September some large inconsistencies are evident. The negative daytime NEE will cause the "observed anthropogenic $CO_2$ enhancement" in equation 8 to be biased high and result in larger monthly scaling factors.

We added the descriptions for clarification on lines 454-464 as "The *a posteriori* results indicate that the annual scaling factors were 1.03 ± 0.10 for 2014 and 1.06 ± 0.09 for 2015. The monthly scaling factors derived from using daytime and all-day observations are also shown in Figure S4. These factors vary seasonally with higher values observed in summer. When using daytime values only, the scaling factors were much larger than the all-day values. This can be seen in Figure 3 by comparing the simulated and observed $CO_2$ mixing ratios. We should note here that the larger scaling factors based on the daytime data could be caused by bias in the *a priori* daily scaling factors used to generate the hourly $CO_2$ emissions (Hu et al., 2018b); the monthly anthropogenic averages; and bias in negative biological $CO_2$ enhancement. Since our study is mainly focused on the seasonality of all-day observations, the monthly scaling factors derived from the all-day approach will be used for the following analyses."

*3. Equations*
*As detailed below, the details of the equations are hard to follow, especially in terms of what is simulated and what is measured. I have made some suggestions to clarify the notation.*
Thank you very much for these suggestions. We have adopted many of these changes. These changes are mainly on lines 175-259. We added subscripts of "obs" and "sim" to represent observations and simulations, respectively.

*Specific Comments*
*L46 change v432 to v4.3.2*
Done as suggested.

*L47-48. "and constrained the anthropogenic CO2 emission categories." This is misleading. The scaling factor approach only constrained the total anthropogenic emissions. The isotopic data were used constrain the cement fraction to some extent.*
Thank you for catching this. We have revised this as "constrained the anthropogenic $CO_2$ emission categories" with "constrained the total anthropogenic $CO_2$ emission"

*L50. "performed well" This is debatable and subjective. The R^2 values for fits to CO2 data were less than 0.2. If you want to comment on WRF performance, please quantify instead of saying "well".*
We changed "performed well in reproducing" with "can generally reproduce".

*L54. Delta_s has not been defined at this point, so you need to say what it means.*
We added "(the mixture of $\delta^{13}C$-$CO_2$ from all regional end-members)" following Delta_s to

define it.

*L58. Change 'plants' to 'plant'*
Done as suggested.

*L78. Change 'by' to 'from'*
Done as suggested.

*L85. I don't think this is the correct IPCC reference. What you want to cite here are the IPCC guidelines on emissions calculations, not the IPCC report on the science of climate change.*
Done as suggested. We changed it to "IPCC (Intergovernmental Panel on Climate Change): 2019 Refinement to the 2006 IPCC Guidelines for National Greenhouse Gas Inventories, available at: https://www.ipcc-nggip.iges.or.jp/ public/2019rf/ (last access: 24 April 2021), 2019"

*L88. Change 'into the inversion of global biological CO2 flux' to 'into the estimation of biological fluxes in atmospheric inversions.' It's not the biological fluxes that are being inverted. Also global is not appropriate here because you are talking about high fossil uncertainties at local scales.*
Done as suggested. We revised it as "These large uncertainties are propagated into the estimation of biological fluxes in atmospheric inversions".

*L122. Change 'have recently be' to 'have recently been'*
Done as suggested.

*L123. Change 'inversion has been' to 'inversions have been'*
Done as suggested.

*L129. Change 'power' to 'the power'*
Done as suggested.

*L151. The NOAA/ESRL lab you refer to should now be referred to as NOAA/GML (NOAA Global Monitoring Laboratory).*
Done as suggested.

*L176. A) does "ms" in CO2_ms refer to measured? Or is the left-hand-side of eq. 1 just a simulated quantity. This seems to be the case from the text above, but to make that clearer, I suggest adopting more intuitive notation such as 'CO2_sim'. B) Delta_CO2 on the right hand side of eq. 1 might be better written as Sum{i=i,n}[Delta_CO2]i to be consistent with eq. 2.*
Here ms refers to simulation. Thank you for pointing this out. Changed as suggested. We have revised the expressions from equation 1 to equation 6.

*L177. Change 'hands' to 'hand sides'*
Done as suggested.

*L179. Again, I think the notation should be clarified. The left hand side of eq. 2, as I understand it, is the simulated value of atmospheric δ13C. Make that notation consistent with that for the simulated value of atmospheric CO2, e.g. δ13C_sim.*
Done as suggested.

*L183. As mentioned above, in eq. 3, instead of Delta_CO2 use Sum{i=i,n}[Delta_CO2]i. Then it becomes very clear what the definition of delta_s is: the enhancement-weighted mean isotopic value of all sources/sinks.*
Done as suggested.

*L184. Change 'the mixture' to 'the enhancement-weighted mean', which is more precise.*
Done as suggested.

*L194-195. I don't agree with: "background air masses should originate from the free atmosphere at heights of 1000 m or higher above the ground". In general there shouldn't be a specific altitude requirement for background air. A more general definition when conducting regional studies would be that background is the concentration or isotope ratio of air when the air enters the regional study domain, which is often determined using back trajectories. The back trajectories (often an ensemble as in the case of STILT) will exit the domain at a variety of altitudes, including possibly below 1000 m. Also, with regard to WLG in particular, this is a remote high altitude site that would be expected to be sampling free tropospheric air most of the time. While it is true that WLG is significantly to the west of the domain 1 border, given the size of the CO2 enhancements (and δ13C depletions) at the observation site, I would expect WLG to be a reasonable background site. As mentioned above, I think a more likely reason why WLG doesn't appear to be a good background site (more so in summer) has less to do with WLG and more to do with potential bias in the dataset. One quick experiment you can do is compare the CO2 values you extracted from CarbonTracker with the CarbonTracker values for WLG. I doubt there will be a substantial difference. If true, this would suggest that WLG should also be a reasonable background for δ13C.*

[Figure]

Figure R7. Comparisons of $CO_2$ background between WLG observation site and Carbon Tracker based model background.

[Figure]

Figure R8. View of WLG site.

Thank you for this comment. We agree that the back trajectories can also originate from below 1000 m, and is illustrated by footprint (Figure 2b, which is defined as the time-weighted values of released back trajectory particles below half of PBLH within each grid cells). Most of the back trajectories originated from the free atmosphere or higher above the ground. For clarification, we deleted the typo of "at height of 1000 m" and added "For example, based on the previous simulation results for the CO2 background sources, most of the back trajectories originate from the free atmosphere or 1000 m higher above the ground (Hu et al., 2019). Further, the footprint at the north/west edge of Domain 1 is relatively small, indicating that most back trajectories were observed above the planetary boundary layer height (hereafter PBLH)." on lines 202-206.

We also compared the $CO_2$ background between WLG site and the Carbon Tracker based model results. This comparison suggests there is ~2 ppm bias in the annual averages. The bias can increase to >5 ppm in the growing season (June-August). One possible reason for this is that the WLG site is dominated by grasslands. The large bias for summer indicates that the WLG site may not be a reasonable choice to define that background $CO_2$ value in summer.

*L198-199. "can cause a high bias in the $\delta$13C-CO2 background when using this approach." Apologies if I am misinterpreting something, but using WLG is much lower (more negative) than the other background approaches. Why 'high bias'?*
Thanks for pointing it out, we revised the typo by changing "high" with "low".

*L200. 'second approach'. It seems that the second approach is a very limited approach whose main purpose is to validate the 'third approach'. Thus, I would move this after the third approach and maybe not call it an 'approach' but say 'in order to test the second approach, we…'.*
The "second approach" applied simulated $\delta_{s\_sim}$ and $CO_2$ enhancement, while the "third approach" only used observed $\delta_{s\_obs}$ and $CO_2$ enhancement, which makes them different.

*L203. Here δ13C_a is referred to as an observed quantity, whereas in eq. 2 the same notation was used to refer a simulated quantity.*
Yes, Here $\delta^{13}C_a$ is referred to as an observed $^{13}C$-$CO_2$, which was used to derive $^{13}C$-$CO_2$ background, we added "Note here that that $\delta^{13}C_a$ represents the observed $\delta^{13}C$-$CO_2$ not the

simulated $\delta^{13}$C-CO$_2$ ($\delta^{13}$C$_{a\_sim}$) as shown in equation 2." for clarification.

*L205-206. 'minimize simulated CO2 enhancement errors' Are you referring to errors coming from NEE here? Is that why you chose the wintertime? If so, state this more explicitly.*
The reason to choose wintertime is to minimize the influence from NEE including the photosynthesis and respiration. The reason to only choose bottom 5% wintertime CO$_2$ observations is to minimize both influence the from ecosystem NEE and anthropogenic CO$_2$ emissions. We revised this sentence as "We defined clean conditions as the bottom 5% wintertime CO$_2$ observations to minimize simulated CO$_2$ enhancement errors from both biological and anthropogenic CO$_2$ simulations on $\delta^{13}$C-CO$_2$ background calculation" for clarification.

*L209. Change 'equations' to 'equation'*
Done as suggested.

*L209. Is Delta_CO2 here derived from CO2_obs – CO2_bg, or is it simulated? The first usage of Delta_CO2 in eq. 1 implies that Delta_CO2 is a simulated quantity, because you write about eq. 1: "CO2 was simulated as the sum of background (CO2_bg) and the contribution from all regional sources/sinks ($\Delta\Delta$CO2)". Perhaps eq. 1 is meant to describe that observed CO2 can be decomposed into a background component and the contribution from all sources. (Thus implying that Delta_CO2 is not simulated using footprints, but rather can be derived from observations and the background estimate.) However, on line 210 you write that in the third approach you do not need to simulated Delta_CO2_i. Please clarify.*
Here the Delta_CO$_2$ is calculated by using CO$_2$ observation minus CO$_2$ background. As described above, we have revised equation 1 and corresponding description.

"Note that $\Delta$CO$_2$ is the sum of all simulated sources/sinks [$\Delta$CO$_{2\_sim}$]$_i$ and represents the total simulated CO$_2$ enhancement. We use $\Delta$CO$_{2\_obs}$ as the observed CO$_2$ total enhancement, which can be calculated by using the CO$_2$ observation minus the CO$_2$ background values." on lines 180-182.

*L213. 'Similar methods…' I wouldn't say the studies referred to use similar approaches. For one, they don't involve isotopes. Second, in most of the approaches referenced, back-trajectories and information from remote sites were combined to determine background. In contrast, and very importantly, the δ13C background determined using method 3 is not independent of the observations themselves. This is a an important point because this is what allows you to define a background that fits so closely to the upper envelope of the observations, despite the fact that the smooth curve fit through the background (Fig. 6) is close to -6 per mil in the summer of 2015. As mentioned earlier, such values are not physically reasonable.*
We agree that these references do not involve isotopes, and they generally applied observations to derive corresponding background for different trace gases. As explained above, In Figure 6, the hourly $\delta^{13}$C background values were derived by combining hourly $\delta^{13}$C observations with the Miller-Tans approach. This derived result is subject to large

fluctuations at the hourly time scale because in equation 4, the derived $\delta^{13}C$ background will have similar hourly variations with the atmospheric $CO_2$ observations. For example, the derived hourly $\delta^{13}C$ background values fluctuated by more than 2‰ within a single day (shown in Figure 6). These large fluctuations are physically unrealistic given our understanding of background values observed at remote sites. For these reasons, we used the smoothing and fitting technique to provide a best estimate of the slow varying background component.

*L221-222. '1000 m above ground' Why wouldn't you use the concentrations from CarbonTracker at the altitudes where the back-trajectories exited the domain, instead of 1000 m agl? This may be a mis-interpretation or mis-reading of the Hu et al. 2019 methodology. As mentioned above, the background concentration should be taken from the lats, lons, and alts at which the ensemble of trajectories exit the domain. These values (500 in the case of STILT?) can then be averaged (and some aspect of their variance, perhaps the std. error of the mean, used to compute an uncertainty.)*

Thanks for pointing out this typo. In the model framework, we choose the latitude and longitude to locate the backward trajectories, and as was found in previous studies most of these trajectories are at altitudes above 1000 m, we deleted "We used the averaged concentration at latitude and longitude when the released particles enter study domain 1" for clarification.

*L223. 'hourly footprint function' say how these were calculated, or provide a reference here. Also, does this imply that footprints and back-trajectories were calculated for every hour of the data record? If so, state this.*

Yes, the variable $\mathbf{\Delta}CO_{2\_sim}$ was derived by multiplying the simulated hourly footprint function with the hourly $CO_2$ fluxes (Hu et al., 2018a; b). Considering the diurnal variations of both anthropogenic and biological $CO_2$ fluxes, 168 footprints were obtained for each simulated hour. This accounted for the back trajectory of particle movement for 168 hours (i.e. 24 hours per day for 7 days) of transport. The 168 footprint will be multiplied by corresponding hourly $CO_2$ flux. We have added these descriptions on lines 240-243.

We revised equation 5 for clarification as

$$\Delta CO_2 = \sum_{i=1}^{168} flux_i \times footprint_i$$

We also revised the description on lines 254-257 as "where $flux_i$ (units: mol m$^{-2}$ s$^{-1}$) corresponds to each $CO_2$ flux category simulated for each domain for a specific hour $i$, and footprint (units: ppm m$^2$ s/μmol) is the model simulated sensitivity of observed $CO_2$ enhancement to flux changes in each pixel. The $i$ contains the hourly footprint during trajectory of particle movement for 168 hours as described above."

*L233. When convolving the fluxes with the footprints to produce Delta_CO2 (here it's usage is clearly as a simulated quantity!), were hourly footprints for a single measurement convolved with hourly NEE to account for covariances in the diurnal patterns of both the footprints and the NEE? In other words, was there just a single footprint per measurement, summed over the*

*7 days, or were there 7x24 footprints saved? When focusing on simulations of biospheric CO2 this factor is very important. I suspect in your case, where anthropogenic fluxes without a significant diurnal cycle are dominant, neglecting this covariance between NEE and transport is reasonable. However, neither eq. 5 nor the text contain any of this information.*

Yes, as answered in the previous comment, there were 7×24=168 footprints saved, which considered the diurnal variations of both biological and anthropogenic $CO_2$ flux. The biological $CO_2$ flux came from Carbon Tracker biological $CO_2$ flux, the anthropogenic $CO_2$ was derived with hourly scaling factors with EDGAR inventories. This method was the same as described in our previous paper (Hu et al., 2018b) in the reference list.

Hu, C., Griffis, T. J., Lee, X., Millet, D. B., Chen, Z., Baker, J. M., and Xiao, K.: Top-Down constraints on anthropogenic $CO_2$ emissions within an agricultural-urban landscape. Journal of Geophysical Research: Atmospheres, 123(9), 4674–4694, https://doi.org/10.1029/2017JD027881, 2018b.

*L251. Here, you say that you used CarbonTracker values above 1000 m, which is different than what you said on line 221, where you implied that you said you used values at 1000 m. Please clarify. However, I'll repeat that there is nothing special about 1000 m. You should use the CO2 value from CarbonTracker at whatever altitude the trajectories left the domain.*

We have revised this typo by changing "on 1000 m" with "above 1000 m". As described above, we tracked the latitudes and longitudes of particles when they entered domain 1. The reason we used the heights above 1000 m, is based on our previous study (Hu et al., 2018a) where we found that most of the released particles are above 1000 m height when entering domain 1.

*L254. Say immediately which version of EDGAR you used.*
Done as suggested, we added v4.3.2 after EDGAR.

*L258. Modify 'the most up to date global inventory' to 'the most up to date global inventory with sectoral detail'.*
Done as suggested, and thanks for pointing it out.

*L264. It is not quite true that EDGAR v4.3.2 is only available for 2010. This is only true for the version with monthly resolution. Please add this qualifier.*
Done as suggested, and thanks for pointing it out.

*L266. Please explain more about how the factor of 1.145 was calculated. For example, was it based just on CT or did it use EDGAR for 2010 and CarbonTracker for 2014 and 2015? The former would be much better for consistency.*
The scaling factor 1.145 was derived by dividing the anthropogenic $CO_2$ in CT in 2014-2015 by in 2010. We revised this sentence as "This scaling factor is based on Carbon Tracker, dividing the same anthropogenic $CO_2$ emissions for YRD in years 2014-2015 by that in 2010." on lines 291-292.

*L269. Change 'posteriori' to 'a posteriori'.*
Done as suggested.

*L275. Again, with notation, please distinguish simulated (or bottom-up) delta_s as in eq. 6 with delta_s determined by a Keeling Plot, e.g.*

$$\sum_{i=1}^{n} \delta_i \times p_i = \delta_{s\_sim}$$

We revised this as, "where $\delta_i$ is the $\boldsymbol{\delta}^{13}$C-CO$_2$ value from source category $i$, and p$_i$ is the corresponding enhancement proportion (i.e. proportions of a specific enhancement i to total CO$_2$ enhancement). We define $\boldsymbol{\delta}_{s\_sim}$ as the simulated carbon isotope ratio of all sources to differentiate it from the observed $\boldsymbol{\delta}_{s\_obs}$".

*L280. How much of non-metallic mineral production is cement? 60%, 90%, 99%? If nearly all, then I recommend stating this here and saying that from here on you will just refer to it as cement.*
Here for the lack of detailed information, we simply attribute 100% of non-metallic mineral production to cement, and added such description on line 308-309.

*L308. Change 'it' to 'them'*
Done as suggested.

*L310. At least for Hu et al., 2019, the inverse modeling approach was very different than that described here. It is a bit misleading to cite it.*
Done as suggested, we deleted Hu et al., 2019.

*L313. Again, confusing notation. Earlier, CO2_obs, was referred to as [CO2]. The "ms" subscript is still a mystery to me. In eq. 1. CO2_ms was total CO2 including all sources (and background). Here, Delta_CO2_ms is only the simulated anthropogenic enhancement. Why not make the subscript more intuitive, such as 'Delta_CO2_anth'?*
Done as suggested, we change $\Delta$CO$_{2\_ms}$ to $\Delta$CO$_{2\_anthro}$.

*L313. Are these terms monthly means? Or is a scaling factor calculated at high time frequency and then averaged to a monthly mean SF? Sargent et al., e.g., calculated the mean of all afternoon modeled and observed enhancements. Also, subsetting to afternoon data only, when modeled PBL heights are likely most reliable, is a necessary test.*
Yes, these terms are monthly means. As described above, the reason to choose all-day averages is that both biological and anthropogenic CO$_2$ flux have strong diurnal variations (i.e. much higher in daytime and lower in nighttime for anthropogenic emissions), so if only use daytime observations, the derived scaling factors will reflect bias in both the *a priori* diurnal scaling factors and monthly averages of CO$_2$ emissions, so even the scaling factors is larger than 1, it does not only indicate the anthropogenic CO$_2$ is underestimated, it can also be caused by underestimation of diurnal scaling factors in daytime not the daily averages.

*L329. As mentioned in the general comments, evaluating model performance with only afternoon data is strongly advised.*

Done as suggested, as answered above, we evaluated model performance in daytime, which seems bias larger than all day performance.

*L331. As mentioned above, I don't agree that R^2 < 0.2 equals "good" performance.*

We revised "The model also performed well in simulating the monthly and seasonal variations of $CO_2$ mixing ratios" as "The model also captured the monthly and seasonal variations of $CO_2$ mixing ratios".

*L348. I'm confused as to how you can compare 2014 and 2015, if 2014 was a full year but 2015 was a partial year. Are you only comparing the months that are common to both years?*

Here as explained on line 157-160, "Further, for an annual comparison, we examined the period from September 2013 to August 2014 (Year 2014) versus September 2014 to August 2015 (Year 2015).", although 2014 and 2015 does not cover the whole 12 months in calendar year 2014 and 2015, they contain whole 12 months spanning two years for further comparison.

*L349. Here you say that the emissions are the same for both years. This should be mentioned explicitly in the methods section.*

Done as suggested, we added "This scaling factor is based on Carbon Tracker, dividing the same anthropogenic $CO_2$ emissions for YRD in years 2014-2015 by that in 2010." in methods section on line 292.

*L353. I don't think you can conclude that meteorological differences between years were relatively small based on comparison of annual values. More analysis would be needed. For example, there could be a lot of seasonal cancellation that still results in similar annual averages.*

Thanks for pointing this out. We revised it as "indicates a relatively small meteorological effect for the annual averages," for clarification.

*L355. In terms of comparing NEE year to year, I think much more analysis is needed for this to be meaningful. First, NEE changes significantly on a seasonal basis. Second, because you are presumably looking at the full record, nighttime respiration is over-represented. Daytime values on the other hand incorporate both daytime NEE (GPP and ER) and respiration from the previous night (and probably more day/night cycles depending on the size of the domain and the windspeeds).*

The hourly $CO_2$ enhancement contributed by NEE is also displayed in Figure 7b, which illustrated obvious diurnal and seasonal variations on lines 510-517.

*L362. Note that the GPP product used is derived from the SIF product used. They are not really that independent. This can be seen in the shape of their seasonal cycles.*

We agree that the GPP is derived from SIF observation and they are not independent.
We added "We note that GPP was derived from SIF, and as a result they share a similar

seasonal cycle." on lines 403-404 for clarification.

*L370. While you may be able to ignore GPP during the winter, it doesn't mean you can ignore respiration. There will still likely be some NEE.*
Yes, we agree that respiration in winter cannot be ignored as displayed in figure 7b. Here in this sentence, we only refer to the comparisons of photosynthesis between growing seasons and non-growing seasons.

*L383. Change Washington to Washington D.C. (to distinguish from Washington state.)*
Done as suggested.

*L385. Change 'Eastern' to 'the eastern'*
Done as suggested.

*L389. "indicating greater…' Most likely, yes, but also trapping of emissions in the PBL will play a role. You cannot immediately transfer enhancements to emissions. So you need to qualify this by saying something like 'assuming similar windspeeds and PBL heights…'*
We deleted the emission comparisons and revised this sentence as "Our enhancements were significantly higher than all of these previous reports of other urban areas"

*L392. Say briefly how the percentages were calculated. I assume by convolution of each regions emissions with the footprints.*
First, we calculated the $CO_2$ enhancement from each province by multiplying emissions in each provincial administrative boundary with corresponding footprint, and then divided the calculated $CO_2$ enhancement by total $CO_2$ enhancement for all area, we added "The $CO_2$ enhancement from each of the 5 zones were simulated by multiplying $CO_2$ emissions in each province with the corresponding footprint." on lines 257-259 in the Method Section for clarification.

*L401. Change 'oil refinery' to 'oil refineries'*
Done as suggested.

*L412. How were the annual scaling factors calculated? As the unweighted (or weighted?) mean of the monthly ones? Also, the MSFs are never presented. What is their seasonality? This seems like a major result of the CO2 part of your analysis (and the 'inversion'). The monthly results should be discussed and/or presented.*
Thank you for this suggestion. The annual scaling factors are the weighted mean of monthly values on lines 453-454 "We constrained the monthly anthropogenic $CO_2$ emissions by using the MSF method (equation 8) and computed the 12-month average to represent the years of 2014 and 2015.". As shown above in Figure R7, the scaling factors when using all-day generally varied around 1, while when only using daytime, they are generally larger than 1. Both results show seasonality. Both of them show obvious seasonality, which is lower from November to March, and higher from April to July and from September to October. We have added this figure in supplemental file and also added "We constrained the monthly

anthropogenic $CO_2$ emissions by using the MSF method (equation 8) and computed the 12-month average to represent the years of 2014 and 2015. The *a posteriori* results indicate that the annual scaling factors were 1.03 ± 0.10 for 2014 and 1.06 ± 0.09 for 2015. The monthly scaling factors derived from using daytime and all-day observations are also shown in Figure S4. These factors vary seasonally with higher values observed in summer. When using daytime values only, the scaling factors were much larger than the all-day values. This can be seen in Figure 3 by comparing the simulated and observed $CO_2$ mixing ratios. We should note here that the larger scaling factors based on the daytime data could be caused by bias in the *a priori* daily scaling factors used to generate the hourly $CO_2$ emissions (Hu et al., 2018b, Figure R9); the monthly anthropogenic averages; and bias in negative biological $CO_2$ enhancement. Since our study is mainly focused on the seasonality of all-day observations, the monthly scaling factors derived from the all-day approach will be used for the following analyses." on lines 453-468.

[Figure]

Figure R9. Diurnal scaling factors in Hu et al. (2018b), Figure 2a.

*L415. Change posteriori to 'a posteriori' and 'for YRD area' to 'for the YRD area'.*
Done as suggested.

*L416. The last sentence on cement seems unnecessary and out of place, even if it is true.*
We deleted the last sentence of "As noted, cement $CO_2$ emissions in the YRD is the largest regional source for global cement production". In 2013, the cement production from China accounted 60% of global total production, the second ranked country is India which accounted for 9% (USGS, 2014), and the YRD area accounted for 20% of national cement production in China, which is 20%*60%=12% in the global production (Xu et al., 2017; Yang et al., 2017). We cited the reference from USGS on line 132.
USGS (U. S. Geological Survey), 2014. Mineral Commodity Summaries 2013.http://minerals.usgs.gov/minerals/pubs/commodity/cement/.

*L429. Using 'discrimination' to describe the isotope ratio of cement production is not appropriate, because it is not a fractionating process like photosynthesis is, e.g. You could instead say 'the isotopic signature associated with cement production…', e.g.*
Done as suggested, we changed "discrimination" with "Enrichment of the isotopic signature".

*L431. Change 'plants' to 'plant'*
Done as suggested.

*L432. Change 'than observed' to 'than the observed'*
Done as suggested.

*L438. Regarding Chen et al., 2006, the vertical gradients in that study are based on models. Observations were generally only at 20 m agl. In at least one example (Fig. 1) The vertical gradient looks to just about 0.1 per mil at mid-day. Moreover, at least for the summer months, the simulated values in the surface layer are more enriched due to photosynthetic drawdown.*
Yes, the model work by Chen et al., 2006 seems $^{13}CO_2$ has small vertical gradient in the daytime because of the well mix of boundary layer, and large gradients of >1‰ was found between 20 m and 500 m in different years, which indicate the distinct signals in boundary layer and free atmosphere.

*L439. Ghasemifard is not in the reference list.*
Thanks for pointing it out, we have added it in the reference list.

*L440. Saying that Zugspitze values were around -7 per mil for winter 2013 is not an accurate characterization of the Ghasemifard results. Looking closely at Fig. 2 of Ghasemifard et al. 2019, the average δ13C for DJF of 2012/2013 is at least 1 per mil lower than -7.0 per mil (even excluding pollution events). This extraordinarily high value of ~ -7.0 is only reached in two cases, once in October and once in January. Also note that -7.0 per mil is an unrealistically high value for Oct. 2012 in relation to similar high altitude data like that from the Scripps Mauna Loa record.*
We revised the sentence as "Recently, Ghasemifard et al. (2019) showed that hourly $\boldsymbol{\delta}^{13}$C-$CO_2$ values at Mount Zugspitze, the highest (2650 m) mountain in Germany, varied between -7‰ and -12‰ in the winter for 2013. During two especially clean air events (in October and February) at Mount Zugspitze, the $\boldsymbol{\delta}^{13}$C-$CO_2$ was approximately -7‰, during which the $CO_2$ mixing ratios varied between 390 and 395 ppm.".

*L442. Saying that a clean air event pulls the δ13C down from -7 to -7.5 does not make sense. Pollution events (δ13C of fossil fuels average around -30 per mil) will make δ13C more negative. Here you are saying a *clean* air event makes the δ13C more negative.*
As described above, we revised this sentence as "During two especially clean air events (in October and February) at Mount Zugspitze, the $\boldsymbol{\delta}^{13}$C-$CO_2$ was approximately -7‰, during which the $CO_2$ mixing ratios varied between 390 and 395 ppm."

*L449. As with CO2, doing an afternoon hours-only analysis of model performance would be valuable.*

[Figure]

Figure R4b. Comparisons of $CO_2$ daily averages for day time.

As described above, we calculated the RMSE, R and MB for daily averages for all-day and daytime, which were 18.68 ppm, 0.44 and 2.68 ppm for all-day averages, and 25.21 ppm, 0.38 and 10.74 ppm for daytime.

[Figure]

Figure R5. Scatter plots of observed versus modeled (a) winter time $CO_2$ mixing ratios, (b) winter time $\delta^{13}C$-$CO_2$, (c) summer time $CO_2$, and (d) summer time $\delta^{13}C$-$CO_2$ for both years, here these dots are day-time(10:-16:00) averages.

We also did the comparisons by only choosing daytime observations, the results indicated daytime $CO_2$ mixing ratio simulations in summer were slightly underestimated. While the simulations in winter can generally capture the trends for both $CO_2$, during which the biological $CO_2$ enhancement played a relatively small role than anthropogenic emissions. This larger bias in summer can potentially be caused by coarse spatial-temporal resolutions in Carbon Tracker biological $CO_2$ flux, which were 1×1 degree with three-hours averages, and urban biological $CO_2$ flux with finer spatial scale and hourly resolution is suggested in following studies. We added this description on lines 533-542.

*L468. 'and generally caused by…' you could test this hypothesis by correlating the model minus obs. residuals for CO2 and δ13C. That is, if δ13C simulation errors are caused by CO2 errors, there should be a strong correlation and the slope should be related to the mean isotopic signature of the sources.*

[Figure]

Figure R10. Relationship of observation minus simulation residual between $CO_2$ and $^{13}CO_2$ for (a) winter in 2013, (b) summer in 2014, (c) winter in 2014, and (d) summer in 2015.

Thanks for this suggestion. We have shown the relationship between observation minus simulation residual for $CO_2$ and $^{13}CO_2$ as displayed in the figure above. Strong negative correlation (with r<0.85) was observed. We have added this figure to the supplemental file and added text "Some large discrepancies are evident and generally caused by the simulated total $CO_2$ enhancement biases (potentially caused by poorly simulated PBLH during these periods) and the negative relationship between $\delta^{13}C$-$CO_2$ and the $CO_2$ enhancement as shown in Figure S6." on lines 525-528.

*L471. In Figure 9, by focusing only on daytime data you may minimize the impact of large night time NEE enhancements (Fig. 7b) and get a better correlation.*

[Figure]

Figure R6. Scatter plots of observed versus modeled (a) winter time $CO_2$ mixing ratios, (b) winter time $\delta^{13}C$-$CO_2$, (c) summer time $CO_2$, and (d) summer time $\delta^{13}C$-$CO_2$ for both years, here these dots are day-time(10:00-16:00) averages.

Done as suggested, we have replied this in above questions.

L477. *After 'was observed in December and July' reference the relevant figure.*
We added "Figure 10a" following "December and July".

L484. *Regarding delta_s, presumably this was calculated using equation 6 and the emission proportions listed on line 397. And are these the all-day values shown in Figure 12? Or are these data derived delta_s values from Keeling Plots. As mentioned earlier, different notation for simulated and data-based delta_s would be helpful.*
The delta_s was simulated $\delta_s$ (as described in equation 6) by using the simulated enhancement proportions. They are both the nighttime and all-day values in Figure b-c. We have revised as $\delta_{s\_sim}$ and added "We define $\delta_{s\_sim}$ as the simulated carbon isotope ratio of all sources to differentiate it from the observed $\delta_{s\_obs}$" on line 303 for clarification.

L496. *The seasonal cycle attribution is very confusing to me. To start with, Fig. 10 shows that about two thirds of the seasonal cycle is from the δ13C background. This seems straightforward. However, then you say that the background and regional source terms are 59% and 41% of the seasonal cycle. And then you say that total CO2 enhancement and CO2 enhancement components further contribute another 20% each. This seems to add up to 140%.*

The regional source term contains both total $CO_2$ enhancement and $CO_2$ enhancement components, so here the contribution of 20% for each is the partition in regional source term, not the added contribution, we rephrased this sentence as "Further, the total $CO_2$ enhancement and change in $\delta_s$, sum of both can be treated as regional source term, contributed equally (about 20%) to the $\delta^{13}C$-$CO_2$ seasonality." on lines 565-566 for clarification.

L499. *I don't think your approach to separately investigate photosynthesis and respiration is valid. Negative NEE instances will still have a substantial influence from respiration. And some positive NEE will have substantial photosynthesis. I highly recommend simplifying the analysis (and make it more accurate) and only analyze the effect of NEE. You just don't have enough information to estimate the effects separately (unless perhaps you use the GPP estimates to partition NEE).*
Thank you for these comments. We have revised this as follows: "(1) excluding negative NEE when photosynthesis is stronger than respiration, and (2) excluding both photosynthetic discrimination and respiration. Note that only NEE was used in our study with no partitioning between photosynthesis and respiration in the daytime. The only role of photosynthetic discrimination should be stronger than in case 1 when only negative NEE is used.". We also changed the caption and label from "without/excluding photosynthesis" to "without/excluding negative NEE" in Figure 10 for clarification.

*L506-507 'via respiration' I would change this to 'via net respiration', or 'positive NEE', because photosynthesis might be active at some points.*
Done as suggested, we changed 'via respiration' to 'via net respiration'.

*L525. Eliminate 'a' prior to 0.40 per mil.*
Done as suggested.

*L526. Why is the impact of cement expressed as a range?*
Here we used both observed $\delta^{13}$C-$CO_2$ seasonality (1.51‰ and simulated $\delta^{13}$C-$CO_2$ seasonality (1.53‰) to subtract the seasonality without non-metallic mineral production sources of 1.47‰. Here, we only use simulated $\delta^{13}$C-$CO_2$ seasonality and revised "0.05‰ to 0.07‰" as "0.07‰"

*L542. Another notation question: what exactly is delta_ms? It has not been defined.*
We changed $\delta_{s\_ms}$ with $\delta_{s\_sim}$ and define it on line 303 as "We define $\delta_{s\_sim}$ as the simulated carbon isotope ratio of all sources to differentiate it from the observed $\delta_{s\_obs}$".

*L561. It's not clear that earlier in the paper you stated a hypothesis.*
We revise this sentence as "These results also indicated that".

*L565. I disagree that the cement isotopic signature is the most distinct. In fact it is only ~ 8 per mil away from the atmospheric value, while biological sources and other fossil sources are ~ 20 per mil away and thus should exert more leverage on the atmospheric value. I also disagree that its emission is large. It is large for cement compared to other parts of the world, but it is still only ~ 10%.*
We revised this sentence as "As discussed above, cement $CO_2$ emissions had the most distinct $\delta^{13}$C-$CO_2$ end-member value of 0‰ ± 0.30‰ when compared with the averages of other anthropogenic sources. Combined with its large emission compared to other regions of the world, it had a strong potential to influence $\delta_s$ and $\delta^{13}$C-$CO_2$" on lines 634-637.

*L567. Provide a reference for the statement that the YRD is the largest cement producing region in the world.*
We added three references to support it as (USGS, 2014; Cai et al., 2015; Yang et al., 2017) on line 638.

*L588. Regarding the 16.85 %, why is there such a large seasonal cycle in the cement enhancement proportion. Can you discuss this or offer any explanation? Seasonal changes in wind direction? Also, I am a bit confused about the a and b superscripts in Table S2. I thought the cement enhancements were simply calculated by convolving the cement emissions and the footprints. Do a and b just refer to whether the proportion is relative to total flux or only anthropogenic flux? If so, I would not use 'considering' in Table S2 and instead say explicitly that a is ratio of cement/anthro and b is a ratio of cement/total.*
The large seasonal cycle of cement enhancement proportion should be driven by source area

changes as discussed "We found a relatively large difference between the enhancement proportion and the emission proportion for oil refineries (from 11.5% to 4.1%) as compared to other categories. This may be because power industry, manufacturing and non-metallic mineral production were more homogeneously distributed compared to oil refineries, which were closer to our $CO_2$ observation site. Further, changes in source footprint caused by wind direction variations likely played an important role." on line 441-446.

We revised the caption in Table S2 as "note the superscript 'a' is ratio of cement to anthropogenic $CO_2$ emissions and 'b' is a ratio of cement to total $CO_2$ emissions, which contains biological and anthropogenic $CO_2$ flux"

*L609. I don't think you can conclude that delta_s is more sensitive to cement emissions than other emission categories without testing those categories!*
We revised it as "These results indicate that $\delta_s$ is sensitive to cement $CO_2$ emissions".

*L616. 'contributed 0.40 per mil' to the seasonal cycle? Please say this explicitly.*
We revised it as "contributed 0.40‰ to the seasonal cycle, accounting for 64.5% in all regional source terms (0.62‰)"

*L631. Change 'write' to 'wrote'*
Done as suggested.

*L665. You shouldn't really cite a 'Discussions' paper from 2016. Please replace with the final paper (or remove if not available).*
Done as suggested, we cited its formally published version.

*L684. The journal title seems to appear twice.*
Thanks for pointing it out, we deleted the first title.

*L845. What is [J]?*
We deleted [J].

*L901. It would help to link panel B to panel A so one can see which of the cement locations are in which domain. Maybe draw some of the domain boxes in panel B?*
Most of the sources came from domain 3, so we draw domain 3 in b.

*L931. Are the units really nmol/m2/s? This implies a \*maximum\* flux of 1e-14 mol/m2/s, based on the colorbar.*
The unite is nmol/m2/s, we have revised this typo.

*L932. The colorbars/legends are very hard to see in the population density maps. Also, panel B colorbar needs units. Also for panel B, which domain in Figure 1 does this concentration footprint correspond to. It's not clear.*
We increase the size of colorbar, we also added the units for panel B in the caption. The footprint corresponds to the same STILT domain setup in (Hu et al., 2019), we added it on

line 268.

*L959. Consider using something besides 'Delta_CO2' as the y-axis label, because this has already been defined differently elsewhere in the paper. Also, in the figure caption be more clear about what 'Delta' means, which (I think) are differences between the same monthly averages in two different years. Panels b,c, and d are referred to after Fig. 5a and are not strongly related to panel a. I recommend separating them from this figure. Also what does 'distance xx' in the legend mean?*

We changed added 'Delta_CO2' as "$CO_2$ enhancement difference," and revised the caption as "(a) Relation between monthly PBL height and change in $CO_2$ mixing ratio, here these dots represent difference of monthly averages in two different years for all hours;" we moved Figures 4b-d after Figure 5, and revised it caption as Time series (2013 to 2015) of (b) NDVI, (c) SIF, and (d) GPP. The distance indicates the radius of area centered with NUIST observation site, and the NDVI, SIF, GPP values are averages in these areas." in the caption of Figure 5.

*L971. In panel a should 'model' really be 'anthropogenic'? b and c are switched in the caption.*

Here the model represented the sum of both anthropogenic and biological $CO_2$ enhancement, we added "note 'model' represents the sum of both anthropogenic and biological $CO_2$ enhancement simulations," on line 1052. We also switched the caption for b and c.

*L1019. Do the individual data points in the plots represent daily means? Please clarify.*

Yes, these dots represent daily means. We added "here these dots are daily averages" on line 1097 for clarification.

*L1051. Relabel panel a y-axis as delta_s.*

Done as suggested.

*Note. There are some language erros/typos in the supplement that need to be fixed*

Done as suggested.